# The composition of British bird communities is associated with long-term garden bird feeding

Kate E. Plummer [1,2], Kate Risely [1], Mike P. Toms [1] & Gavin M. Siriwardena [1]

There is a multi-billion dollar global industry dedicated to feeding wild birds in residential gardens. This extraordinary boost to food resources is almost certainly reshaping entire bird communities, yet the large-scale, long-term impacts on community ecology remain unknown. Here we reveal a 40-year transformation of the bird communities using garden bird feeders in Britain, and provide evidence to suggest how this may have contributed to national-scale population changes. We find that increases in bird diversity at feeders are associated with increasing community evenness, as species previously rarely observed in gardens have increasingly exploited the growing variety of foods on offer over time. Urban areas of Britain are consequently nurturing growing populations of feeder-using bird species, while the populations of species that do not use feeders remain unchanged. Our findings illustrate the on-going, gross impact people can have on bird community structure across large spatial scales.

[1] British Trust for Ornithology, The Nunnery, Thetford, Norfolk IP24 2PU, UK. [2] Centre for Ecology and Conservation, College of Life and Environmental Sciences, University of Exeter, Penryn Campus, Cornwall TR10 9FE, UK. Correspondence and requests for materials should be addressed to K.E.P. (email: Kate.Plummer@bto.org)

Food availability may be the single most important factor determining the size and distribution of animal populations. Winter food availability, in particular, plays a critical role in regulating bird populations in seasonal environments[1], with over-winter survival a key cause of population change for many terrestrial bird species[2–4]. In much of Europe and North America, the deliberate provision of food in domestic gardens and yards (garden bird feeding) has modified the winter resource base for birds extensively[5], helping to buffer against environmental drivers that operate through changes in natural food abundance[6]. In Britain, for example, homeowners are reportedly providing enough supplementary food to support approximately 196 million birds, far exceeding the combined, total population of many common garden species[7]. This massive human intervention is likely to be having profound repercussions on the bird communities around us[5].

Early feeding pioneers attracted a relatively simple bird community to gardens using kitchen scraps and home-made table feeders[5]. However, the practice has changed substantially since becoming commercialised in the mid-20th century[8]. It now seemingly benefits a much broader bird community, although it is unclear whether or not some species may have lost out due to heightened interspecific competition for access to supplementary foods[9,10]. Previous research has demonstrated that the distribution of feeders across a city can predict avian abundance patterns for some species[11], with bird community composition also reacting promptly to the introduction and removal of feeding stations[12]. This would suggest that garden bird feeding is capable of supporting local populations and enhancing bird numbers, at least in the short-term. The availability and nutritional value of anthropogenic foods are also likely to have substantial impacts (both positive and negative) on the health, survival and breeding outputs of wild birds[3,13,14]. But ultimately, how these effects scale up to influence bird community dynamics and population trajectories across entire landscapes is still unknown.

In Britain, gardens cumulatively account for approximately one quarter of all urban land cover[15] and play an important role in supporting the national populations of many bird species[16–18]. Given our awareness that at least half of British homeowners feed the birds in their gardens[7,19], and our growing understanding of the extensive ecological and evolutionary impacts of supplementary food use, it is reasonable to predict that garden bird feeding might also be influencing bird communities across large spatial scales. The supply and uptake of garden bird food during winter has been monitored throughout Britain since the 1970s, via the Garden Bird Feeding Survey (GBFS, Supplementary Fig. 1), providing a unique opportunity to study long-term shifts in bird communities in response to large-scale food supplementation. Here, first, we characterise real-time growth and innovation within the garden bird feeding industry. Then, using data extracted over a 40-year time series, we demonstrate that, over time, food resources provided by the British public have altered the composition of bird communities utilising garden bird feeders and, consequently, have helped to shape the national populations of birds in Britain today.

## Results and Discussion
**Garden bird feeding industry.** To quantify long-term changes to garden bird feeding in Britain, we conducted a comprehensive review of advertising in Birds—a charity membership magazine reaching more than 2 million readers[20]—published between 1973 and 2005. By definition, the magazine's audience have a keen interest in birds and their conservation, thus representing the target market for companies selling popular and novel feeding products. Since brand advertising is known to impact total industry

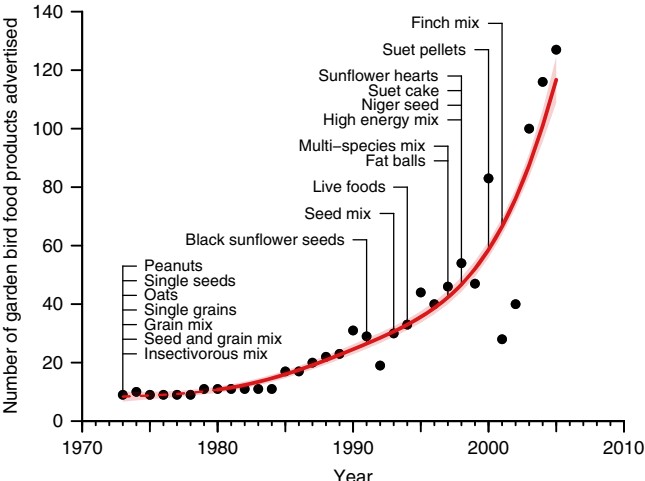

**Fig. 1** Changes to the British garden bird feeding industry over time. Temporal trend in the number of unique garden bird foods advertised between 1973 and 2005 in RSPB Birds magazine with 95% confidence limits (light shading). The period of significant change is shown with a solid line, and the non-significant period (up to 1980) with a dashed line. Text labels indicate to the years in which different broad food types were first advertised

demand[21], advertising patterns in Birds are expected to provide meaningful indices of consumers' garden bird feeding habits.

The proportion of advertising space dedicated to the bird feeding industry increased significantly over time ($\chi^2_1 = 26.19$, df = 2.62, $p < 0.001$; Supplementary Fig. 2), indicating the rising popularity of feeding wild birds. After accounting for changes to advertising practices, we revealed upward trajectories in the total numbers of food ($F = 178.1$, df = 3.15, $p < 0.001$) and feeder ($F = 187.9$, df = 3.70, $p < 0.001$) products on offer, including exponential increases from 1980 onwards (Fig. 1; Supplementary Fig. 2). The number of food types available ($\chi^2_1 = 8.06$, $p = 0.005$) and their diversity (Simpson's Index, $\chi^2_1 = 7.26$, $p = 0.007$) also grew significantly (Supplementary Table 1; Supplementary Fig. 2). Specialist foods, such as sunflower hearts and fat balls, first appeared in the 1990s (Fig. 1) as companies, supported by conservation bodies, deliberately diversified their products to attract more species[5]. This broadening of food resources, added to the potential for selective provisioning by homeowners, may have both influenced and complicated bird community responses to increases in food quantity.

**Community changes at garden bird feeders.** Using 40 years of GBFS data from 1973/74 to 2012/13, we identified 133 bird species using garden feeders during winter, equating to 52.6% of all species, excluding vagrants, found in Britain[22] (Supplementary Table 2). We analysed the long-term trends in community composition, nationally (using a single, rarefied time-series) and within individual gardens (using mixed effects modelling), to examine evidence of bird community adaptation in response to evolving feeding practices (see Methods section).

Across Britain, there has been a highly significant increase in the diversity of birds using garden bird feeders since the 1970s, according to Simpson's Diversity Index ($F = 355.0$, df = 3.52, $p < 0.001$; Fig. 2a). Although, nationally, the same suite of species have continued to use feeders over time ($\chi^2_1 = 0.00$, $p = 0.99$; Fig. 2c), homeowners are encountering an increasingly species-rich ($F = 143.5$, df = 2.99, $p < 0.001$; Fig. 2d) and diverse ($F = 123.0$, df = 2.92, $p < 0.001$; Fig. 2b) community of birds visiting the feeders in their individual gardens.

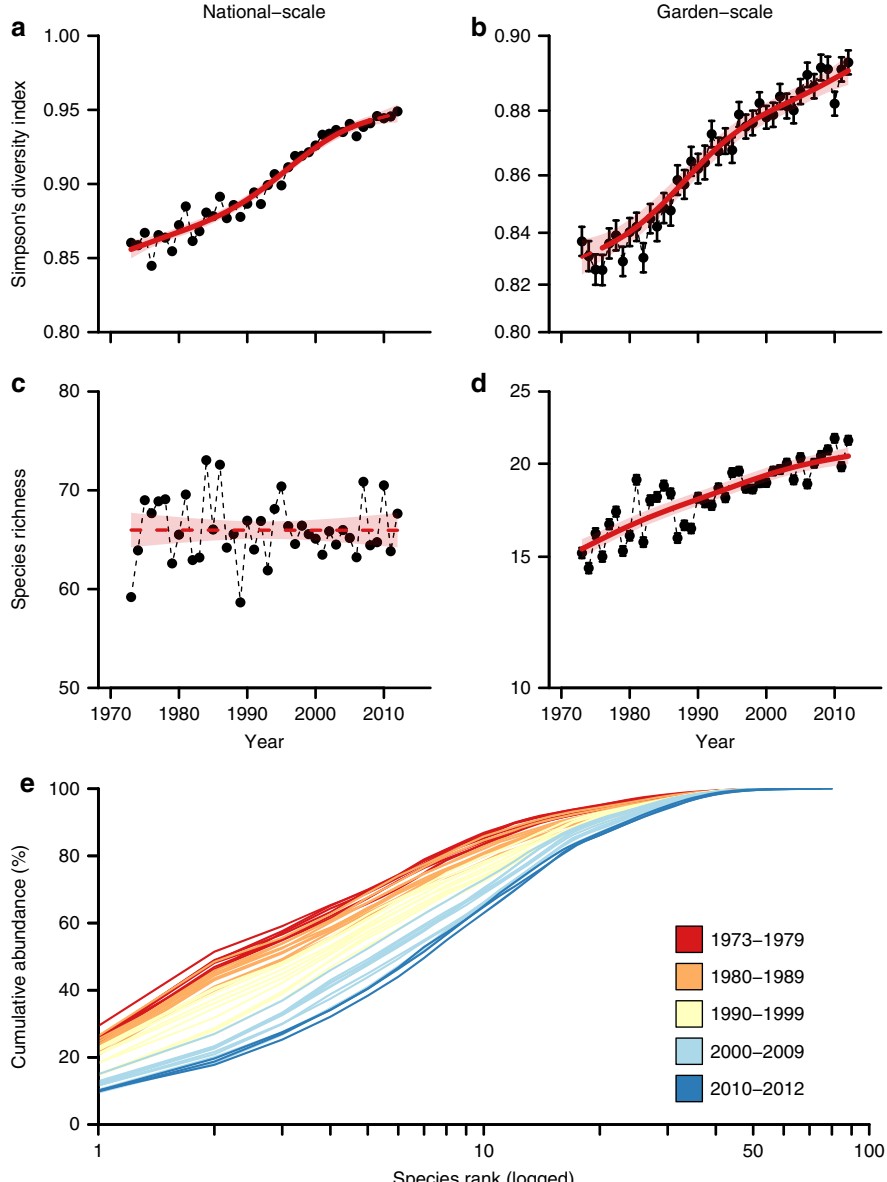

**Fig. 2** Long-term changes in the diversity of birds using garden bird feeders. **a**, **b** trends in Simpson's diversity at a national-scale (**a**), and a garden-scale (**b**). **c**, **d** trends in species richness at a national-scale (**c**), and a garden-scale (**d**). For **a**–**d**, model-fitted trends are shown with 95% confidence limits (pale shading), periods of significant change are shown with a solid line and non-significant periods with a dashed line; dots show annual values (**a**, **c**) or annual means with error bars depicting s.e.m. (**b**, **d**). **e** k-dominance curves for each year in the time-series ($n = 40$) coloured by decade

This combination of large- (national-) and small- (garden-) scale patterns suggests that many species have become more abundant at garden feeders over time, potentially resulting in a change in species dominance within the feeder-using community. Indeed, a comparison of k-dominance curves from each year revealed a clear pattern of increasing evenness over time (Fig. 2e). Most notably for example, approximately half of all birds using feeders belonged to just two species in the 1970s, but by the 2010s, the number of species making up the same proportion of the community had more than tripled. We examined the possibility that this finding might purely reflect the changing spatial ranges of British birds[23], potentially influencing community trends by bringing more wintering bird species in contact with monitored feeders over time. However, the median net change in the proportion of GBFS gardens located within a species' range between 1981 and 2011 was only 2.49% ($n = 130$)[24,25]. This suggests that garden bird feeders, specifically,

could be responsible for attracting more individuals across a greater species range as time has passed.

We used GBFS data on the numbers of hanging, table and ground feeding units (collectively termed feeders) to evaluate the importance of garden bird food availability in driving community patterns directly. As expected, individual homeowners supplied an increasing number of feeders ($F_{1,6431} = 195.6$, $p < 0.001$), particularly hanging feeders ($F_{1,6431} = 297.2$, $p < 0.001$), over time (Supplementary Fig. 3). It is reasonable to assume that this increase in feeder numbers also reflects the greater variety in food types that became available during the same timeframe, since homeowners are likely to aim to attract more species by diversifying food provision, rather than simply increasing provision of the same foods.

Changes in bird communities across the British countryside have previously been shown to be at least partially linked to climate change and urbanisation[26,27]. Indeed, variation in garden use by birds is also known to be associated with winter

temperatures and local habitat characteristics[18,28]. But interestingly, when we included all of these potential drivers as covariates in the modelling of bird community temporal trends, we found that the number of feeders provided in a garden had a greater influence on species richness and diversity than either winter temperature or local habitat (Supplementary Fig. 4). It therefore seems that the broad temporal changes observed at garden bird feeders could be the result of the cumulative changes in food provisioning across multiple gardens, allowing species rarely observed at the start of the time-series to take better advantage of feeders over time.

Well-defined interspecific dominance hierarchies are known to exist at bird feeders, as species of different sizes and competitive strengths fight for to access food supplements[9,29]. Our findings suggest that, by increasing the number of feeders available in gardens, this has reduced the capacity for resource monopolisation by any one species. Concurrent increases in the diversity of food and feeder types on offer are also likely to have led to greater opportunities for species with more specialist foraging requirements. To this end, the continuing modifications made by homeowners to their feeding practices appear to have contributed significantly to the changing composition of bird communities in gardens.

**Links to national population change**. It would seem that the composition of bird communities exploiting garden bird food has changed in parallel with evolving feeding practices. More generally, community changes in terrestrial birds are likely to be the product of many different factors, including climate, habitat change, resource availability, conditions in breeding/wintering grounds or on migration, disease, competition and predation. Indeed, as previously mentioned, wider community changes have occurred across Britain[27], and therefore it is feasible that apparent increases in feeder use could merely reflect the greater detection of birds whose populations have grown through a mechanism unrelated to garden bird feeding. So, are changing feeder communities simply reflecting these wider patterns[18], or could feeding actually be a driver of population change?

To answer this, first, we needed evidence that there is a real association between the use of garden bird feeders during winter and concurrent changes in species' population sizes. Species that regularly visit garden feeders are most likely to experience population-level impacts of supplementary food use. Therefore, we identified 39 species that regularly visit garden feeders (feeder-users) and tested whether independent estimates of their national breeding population trends, derived from Massimino et al.[30], could be linked to shifting winter feeder usage. Feeder use—the proportion of gardens where each species used feeders—increased by an average of 14.9% (s.e.m. 4.0%) between 1973 and 2012, with two thirds of species showing a significant positive change (Fig. 3a; Supplementary Table 3). Further, these changes were found to be significantly associated with national population changes over the same timeframe ($r^2 = 0.43$, $F_{1,29} = 21.82$, $p < 0.001$; Fig. 3b). We accounted for phylogeny in our analysis, under the assumption that more closely related species would be more similar in their tendency to use feeders and in the extent to which their populations change. Indeed, the estimate of Pagel's lambda (a measure of the phylogenetic signal) for species population change was significantly different from zero ($\lambda = 0.94$, $p = 0.014$). However, there was no evidence that feeder use was influenced by phylogeny ($\lambda < 0.001$, $p = 0.15$), and the optimised lambda value from the regression model ($\lambda < 0.001$) denotes very limited covariance between feeder use and species' population change (distinct from the explanatory power of feeder use per se). Feeder use is typically associated with passerine

birds[5], but, with use of supplementary food evident across the phylogenetic tree (Supplementary Fig. 5), these findings give a strong indication that the proliferation of feeder use within bird communities is independent of species' inter-relatedness.

Previous research has demonstrated that, across species, bird population trends are significantly reduced in urban areas compared to other habitats[31]. Therefore, to understand whether feeding, specifically, could be contributing to the wider population changes reported, we needed to be able to separate the influence of feeder use from other co-varying drivers of garden bird numbers that also operate along a gradient of urbanisation[32]. To achieve this, we focused our analysis on the changes in bird populations occuring in urban areas of Britain only, where garden bird feeders are widely accessible to all birds[19], allowing us to test the effect of feeding independently of any other potential influences associated with the urban gradient[32]. Using all available trends for species ($n = 72$) occurring in urban areas of Britain (namely, all urban, suburban and rural human-dominated habitats[31]) we tested for a difference in the urban population trends for the species that regularly use garden feeders ($n = 33$), compared to those which do not ($n = 39$). Compellingly, feeder-users had significantly different population trajectories from those species with equal access to, but which do not commonly visit, feeders ($F_{1,70} = 7.43$, $p = 0.008$; Fig. 3c), with no phylogenetic influence (Pagel's $\lambda < 0.001$). In fact, while there was no overall directional change for the species that do not use feeders, by comparison populations of feeder-users increased significantly (Fig. 3c).

Since these results do not distinguish cause and effect between increasing supplementary food use and population growth definitively, we checked whether or not similar differences between the two species groups also occured throughout the rest of Britain (i.e. non-urban areas). Finding the same pattern would suggest that the differences observed in urban areas are reflecting broader population changes, with birds moving into gardens in approximate proportion to their availability by area, following something like an ideal free distribution across the wider countryside[33]. However, we found no difference between the non-urban trends for these groups ($F_{1,70} = 1.71$, $p = 0.196$), implying that the relationship is more likely to have resulted from birds choosing to use garden feeders than from them expanding their habitat use due to population growth. While many other factors are certain to influence interspecific variation in trends, these findings provide the first landscape scale evidence consistent with garden bird feeding having influenced population change.

Wild bird feeding has become engrained into human culture across many areas of the world within the last half-century, to the extend that this seemingly small-scale activity is now frequently acknowledged for its demonstrable benefits to human well-being[5,34–37]. Nonetheless, the historical basis for feeding wild birds began with the perception that, by providing food during winter, one can improve the survival prospects of vulnerable individual birds[5]. Changes to bird feeding activities conducted in gardens are already reported to have resulted in stark ecological and evolutionary responses within some individual bird species[38,39]. Our findings indicate that the consequences of feeding reach further still, with evidence that this habitual human activity is also associated with the national-scale restructuring of bird communities.

Intuitively, the types of food provided should affect the types of species attracted[10,12]. Our results indicate that the diversifying commercial bird food market has enabled a growing number of species to exploit supplementary foods over time, while some appear to have lost out as a result of behaviourally dominant or better adapted species becoming more common within the community. Indeed, the bird assemblage that commonly uses feeders (Fig. 3a) includes species of high conservation concern[30],

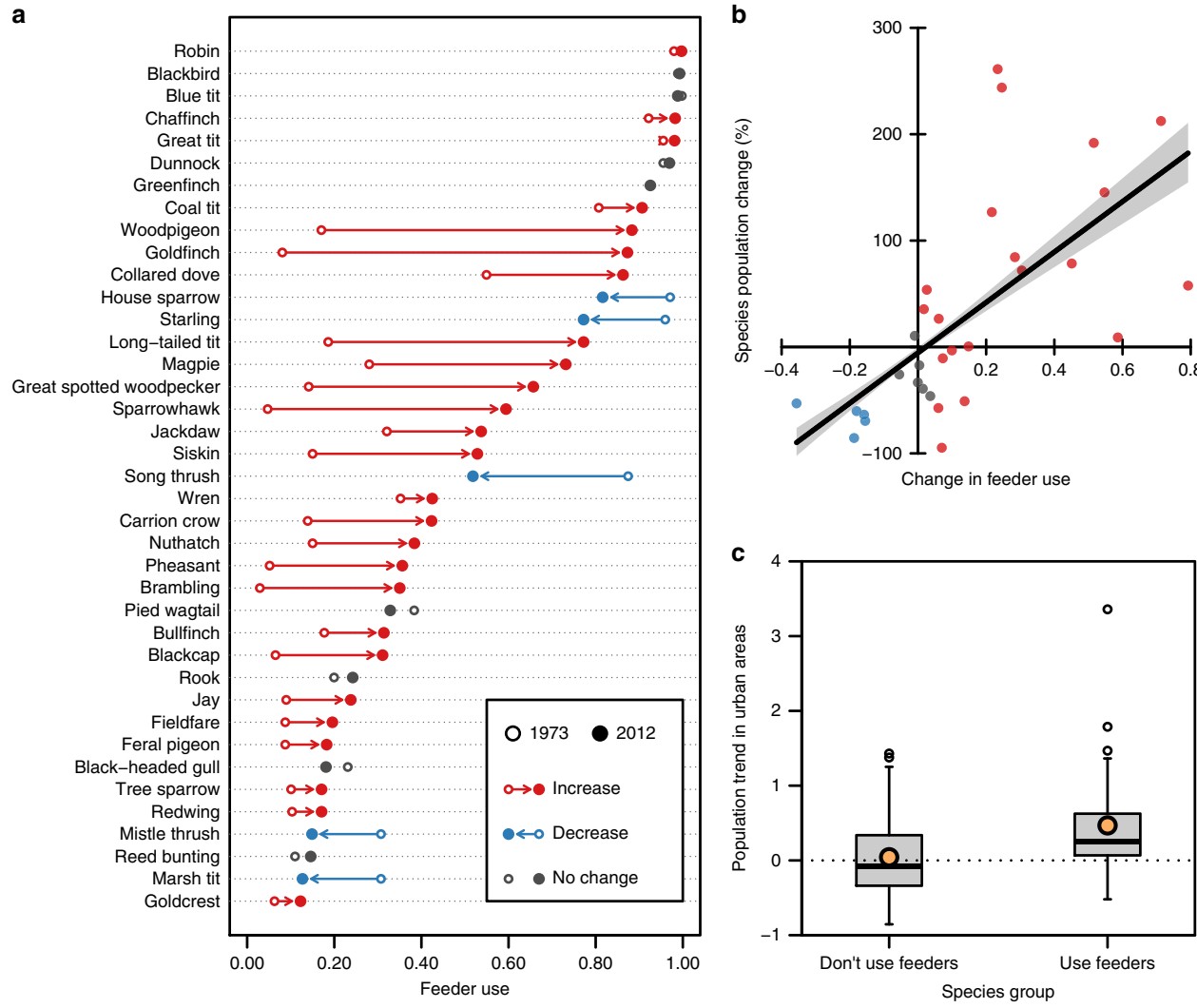

**Fig. 3** Effects of garden feeder use on population change. **a** Change in feeder use (proportion of gardens where species used feeders) by 39 species between 1973 and 2012. **b** The relationship between the change in feeder use and species national population change over the same timeframe with 95% confidence limits ($n = 31$ species). Colours, carried over from panel **a**, depict significance of food use changes. **c** Box plot for the difference in urban population trends for species that do ($n = 33$) or do not ($n = 39$) use garden bird feeders. Boxes bound the interquartile range (IQR) divided by the median, whiskers extend to $1.5 \times$ IQR beyond the boxes and orange dots indicate group means

species capable of promoting human well-being[40] and species considered common pests[41]. Feeding is, therefore, highly likely to have already had important effects, and greater coordination of feeding activities, across networks of gardens and at multiple spatial scales, could be an innovative way of delivering large-scale conservation or species management outcomes in the future[42].

Feeding birds is a growing practice throughout the world, with many people shifting from traditional, winter-only feeding to provisioning all year round[5]. If feeding continues to intensify, it will likely exacerbate the species- and community-level consequences observed here. Greater food diversity, innovation in feeder design, variation in food quality and behavioural adaptation by birds all have the potential to influence the frequency of feeder use and the benefits or detrimental impacts accrued, with downstream consequences for population sizes and community structure. The positive influences of feeder use on population size reported here are likely to be the product of a combination of improved survival[3,4], better physiological condition[13,43] and increased productivity[44] among the individuals frequenting feeders. However, negative impacts of supplementary feeding have been widely reported, particularly those associated

with increased disease transmission at feeders and the poor nutritional quality of food supplements[45–48]. Further research is needed to determine whether, and how, these might limit population growth. Individual decisions by homeowners to feed wild birds can impact cumulatively upon bird communities across large spatial scales. As such this growing, global phenomenon has profound potential to influence biodiversity further and should not be underestimated.

## Methods

**Evidence of garden bird feeding industry change**. Garden bird feeding industry data were derived from advertisements featured in Birds magazine; a free publication, widely distributed to members of the Royal Society for the Protection of Birds (RSPB; recent membership figures totalled over 1.2 million[49]). Our assessment focused on all advertisements promoting garden bird food and/or feeder products in the Autumn editions of Birds from 1973 to 2005. After 2005, advertising was biased toward RSPB-branded products. Over the 33-year period considered, the number of pages featuring all forms of advertising in Birds increased linearly (GLS $\chi_1^2 = 26.19$, $p < 0.001$), correlated with a general increase in magazine length ($r = 0.901$, $n = 33$, $p < 0.001$). For every advertisement ($n = 179$), we extracted data for its size (proportion of page covered), and the names and descriptions of individual food and feeder products. Foods were also allocated to

one of 21 different food type categories (see Supplementary Table 1) using product descriptions, images and online information.

To test for temporal changes in supplementary food quantites, the food and feeder product ranges from all advertisements were summed (respectively) per magazine ($n = 33$). To test for temporal changes in supplementary food diversity, we quantified the number and diversity (using Simpson's diversity index) of food types represented in each magazine. We ruled out the possibility that observed patterns were confounded by increased advertising space by controlling for the total number of pages featuring bird feeding industry advertisements (advertising pages). Number of bird food products and number of feeder products were modelled seperately as a function of a year smooth using generalised linear models (GAMs), with a Poisson error distribution. Advertising pages was also log-transformed and included as an offset, since the numbers of products advertised was expected to be proportitial the the total amount fo advertising space available. The number of different food types and food diversity over time were modelled as a function of year using generalised least squares (GLS) with advertising pages included as a covariate. Food type number was log-transformed to reach normality of the residuals. In a further test for overall growth in the bird food industry, we modelled the proportion of total advertising space dedicated to feeding products as a function of a year smooth using a GAM. Full details about model specifications are given below.

**Garden Bird Feeding Survey**. Data from the Garden Bird Feeding Survey (GBFS) were used to investigate changes in bird communities at feeders in mainland Britain between winter 1973/4 and winter 2012/13 (40 years). GBFS is an annual survey monitoring the number of birds visiting feeders in domestic gardens over a 26-week period from October to March (www.bto.org/gbfs). The survey comprises an average of 217.8 (s.e.m. 7.1) gardens each year, covering a representative range of garden types (suburban/urban and rural) and a consistent geographic distribution. Participants leaving the scheme are replaced with new volunteers from the same region, and with gardens of a similar type and size.

Survey participants record the maximum number of each bird species observed simultaneously using feeders (i.e. at/on feeders and in their vicinity) each week or, in the case of sparrowhawk, feeding on birds using feeders. Data for all species known to occur in Britain according to the British Bird List[22], except vagrants ($n = 6$ species), were used to estimate community indices ($n = 133$; Supplementary Table 2). Scarce migrants, summer migrants (recorded at the beginning or end of the winter period) and species not traditionally considered as garden visitors (e.g. wetland birds) were retained to avoid removing evidence of community change. Records for domestic and aviary species were excluded ($n = 21$). Species with ambiguous identification, such as marsh tit and willow tit, were combined. Changes in community composition are unlikely to have been biased by long-term changes in the arrival and departure of British breeding migrants, since estimated phenological shifts are not large enough to have noticibly increased the probablity of these migrants being recorded by GBFS[50]. Similarly, although volunteer field-skills are not formally controlled, there is no reason to suspect spatial biases or temporal change in the accuracy of species' identification and counts.

Since diversity estimates are influenced by sampling effort[51], the data were restricted to gardens with at least 20 weekly submissions for a given winter (mean = 25.13 weeks), as this number of replicates was seen to produce reliable species richness and species-specific abundance measures. More specifically, species accumulation curves, averaged across gardens each winter, reached an asymptote with 20 weeks of surveying. Further, we used general linear mixed models (GLMMs) controlling for garden and year random effects to verify that increasing sampling effort over 20 weeks had no effect on winter abundance (defined as the maximum count observed in a garden across all weeks surveyed). Winter abundance was independent of sampling effort in 94% of species ($n = 125/133$), significantly more than expected by chance ($z$-test, $\chi^2 = 101.2$, $p < 0.001$, 95% CI = 89.2 – 100.0%). We note that maximum counts provide an accurate (asymptotic) means of comparing changes in relative abundances across species, but probably under-represent true abundance.

The filtered data set used to conduct the analyses comprised 1,001 GBFS gardens in mainland Britain (mean ± s.e.m. per year = 185.8 ± 6.1) that contributed a total of 186,825 weekly submissions over the 40-year survey period (Supplementary Fig. 1). We do not expect the data set to contain any biases that might influence the observed findings, given that the data were collected across a consistent set of gardens using structured sampling to a defined protocol, and that the species list has been carefully validated and sampling effort controlled. Any erroneous data, for example due to species misidentification, should be a source of noise rather than bias.

**Potential drivers of bird community change**. To estimate the potential for species' spatial range shifts to impact measures of community temporal change, distribution data from the 1981/82–1983/84 Winter Atlas[25] were compared with equivalent data from Bird Atlas 2007–2011[24] to identify areas where a species had apparently colonised or apparently disappeared at a 10-km resolution. Data were available for 98% ($n = 130$) of all species represented in the GBFS data set (Supplementary Table 2); three species were not recorded during either winter atlas period. GBFS gardens were assigned data from the 10-km squares in which they were located, allowing the net change in the number of GBFS gardens located

within a species' range to be calculated and averaged across all species (using the median due to data skew).

In addition to examining bird community changes through time, three potential drivers of garden bird feeder use were also considered: number of feeders, winter temperature[18] and local habitat[28]. The numbers of hanging feeders, bird tables and ground feeding stations (collectively termed feeders) were recorded each week in surveyed gardens. Weekly feeder numbers were averaged annually by feeder type and in total, to provide an indicator of supplementary food availability throughout winter in each garden. To examine changes in the provision of garden bird feeders over time, numbers of feeders, in total and by feeder type, were log-transformed and fitted separately against year using linear mixed models (LMM) with garden identity included as a random effect. The log-linear functional relationship, which specifies the percentage change in numbers of feeders over time, was applied to achieve normality of the residuals.

Gardens were classified as either suburban/urban or rural according to their surrounding habitat. Suburban/urban includes gardens in areas with a mix of built cover and green space, or in dense urban areas with little vegetation, such as town centres. Rural includes gardens in areas away from towns, with just a few scattered houses, farms or other isolated buildings. The difference in garden habitat types was verified using Land Cover Map 1990[52], which showed that rural gardens were located in 1-km squares with significantly less urban cover on average than suburban/urban gardens (rural = 12.04% ± 0.70 s.e.m.; suburban/urban = 42.66% ± 1.06; $\chi^2 = 40265$, $p < 0.001$). Gardens were re-classified from rural to suburban/urban if urban encroachment occurred ($n = 6$ gardens), but garden identity remained unchanged.

We expected that weather conditions throughout the whole bird data collection period would have a greater influence feeder use than extreme weather events. Therefore, annual measures of average winter temperature were estimated using mean monthly temperature for October – March and used to test for climatic effects on feeder use. Mean monthly temperature (°C) data were extracted from the UK Meteorological Office Climate Projections (UKCP09) 5 × 5-km resolution gridded data set[53]. Gardens were assigned averaged winter temperature data for the 5-km square in which they were located.

**Evidence of bird community change**. We used annual measures of species richness, Simpson's diversity index and $k$-dominance to examine bird community patterns. Species richness was the total number of species observed throughout the winter. Simpson's diversity index was used to provide a robust and meaningful measure of community diversity per winter[51]. Since Simpson's diversity incorporates both the number of species present and their relative abundances, its comparison with species richness was also used to infer changes in bird community evenness. $k$-dominance curves—which plot the cumulative abundances of all species in a community (as percentages) against their species rank (logged)—were used to study changes in community evenness over time[51]. We used species abundance and rank, averaged annually across gardens, to compare $k$-dominance curves from each year of the time-series. The higher the curve, the less diverse and more uneven the community it represented.

To estimate national indices of species richness and Simpson's diversity, we compiled data from all gardens into a single time-series, then applied sample-based rarefaction to standardise sampling effort through time[54]. Specifically, 115 gardens (equivalent to the minimum number surveyed in a single year) were randomly resampled without replacement from the total pool of gardens surveyed per year to achieve a consistent sample size over time. For each year, data from all resampled gardens were pooled and species richness and Simpson's diversity calculated. Resampling was repeated for 1000 iterations and diversity measures averaged. Confidence intervals were not generated, since estimates derived from rarefaction are dependent on the size of the subsample and are therefore not informative about sample variability. The rarefied measures of Simpson's diversity and species richness were modelled separately to quantify national-scale bird community trends. Simpson's diversity was fitted against a year smooth using a GAM, and species richness was fitted against year using GLS regression.

To assess bird community trends at the garden scale, annual measures of species richness and Simpson's diversity per garden were fitted using generalised additive mixed models (GAMMs) with garden identity included as a random term ($n = 7433$ garden-years). To evaluate the influence of other garden use drivers on bird community change, we also included number of feeders, winter temperature and habitat as fixed effects. These terms were standardised to a mean of 0 and s.d. of 0.5 to enable effect sizes to be compared directly[55].

**Linking feeder use to national population change**. Since feeder use would need to be reasonably prevalent within a population to incur national-scale impacts, we focused on species that regularly used feeders when testing for associations between changing feeder use and changes to population size. Data were combined across all gardens per year to derive a single, intuitive index of overall feeder use per species in Britain, defined as the proportion of GBFS gardens in which a species was observed using feeders. Using site occupancy to derive feeder use, as opposed to species abundance, produces an easily interpreted measure of the scale of feeder use nationally, while also minimising the influence of stochastic variation in species counts. We conservatively defined species that regularly used feeders (feeder-users) as

those with a mean feeder use of ≥ 0.1 (i.e., observed in an average of 10% of surveyed gardens per year across the study period; $n = 39$ species; Supplementary Table 3).

For all feeder-users, a binomial generalised linear model, testing the difference in feeder use between the first and last three years in the time-series (1973/4–1975/6 vs. 2010/11–2012/13), was used to estimate the value and significance of net change in feeder use. This approach was used as it could produce a measure of change that was analogous with estimates of national breeding population change over the same timeframe, while also minimising any influence of inter-annual stochasticity and avoiding assumptions about the shape of the temporal trend. More specifically, breeding population changes for feeder-users were similarly calculated as the difference between smoothed annual indices for 1974 and 2013 (i.e. the breeding seasons immediately following the beginning and end of the time-series), derived from the joint Common Bird Census/Breeding Bird Survey (CBC/BBS) trends for England[30]. CBC and BBS use structured, stratified protocols to monitor national bird populations and inform the UK Biodiversity Indicators. Trends were not available for eight feeder-user species, which included winter migrants and species not well covered by the CBC.

Phylogenetic generalised least squares regression (PGLS) was used to test the relationship between changes in feeder use and national population size ($n = 31$ species) while accounting for phylogenetic structure. Bird phylogeny was based on a pruned consensus tree produced by majority rules using 100 phylogenetic trees randomly extracted from the avian phylogenies developed by Jetz et al.[56] (Supplementary Fig. 5). Within the pruned tree, Eurasian nuthatch (Sitta europaea) was represented by phylogenetically similar white-tailed nuthatch (S. himalayensis)[57] and lesser redpoll (Carduelis cabaret) by common redpoll (C. flammea)[58] since these species were absent from the global avian tree. Maximum likelihood was used to estimate the PGLS model's Pagel's lambda, giving a measure of the phylogenetic covariation between the predictor and response. Pagel's lambda values of zero indicate that the predictor-response relationship is unrelated to phylogeny, whereas high lambda values indicate a strong similarity in the relationship between closely related species. To ensure that the error associated with each annual index value was accounted for within the final model outcome, we used a bootstrap procedure to produce 95% confidence limits around the PGLS regression line. For each bootstrap sample ($n = 1000$), new values of feeder use and population index for the beginning and end of the time series were drawn at random from the confidence limits around their original estimates, and then used to recalculate estimates of change. The PGLS model was fitted to each bootstrap sample with lambda set at the value estimated for the original model, then 95% confidence limits were calculated from the set of regression coefficients produced.

PGLS, using the pruned consensus tree and maximum likelihood to estimate Pagel's lambda, was also used to test for differences in population trends between feeder-users and non-feeder users. Here, we used the 1994-2012 habitat-specific trends from Sullivan et al.[31] for all breeding birds that are associated with urban areas of Britain and therefore have frequent access to garden bird feeders. These trends were available for 72 species, 33 (46%) of which had been defined as feeder-users. Feeder-users without trends were either winter migrants or did not have suitable data for population estimation. We aggregated the trends for 12 individual habitats to derive two broader trends of interest, urban and non-urban. More specifically, urban trends were estimated using a weighted average of the trends for suburban/urban settlements and rural settlements, accounting for their habitat availability. Non-urban trends were estimated using an weighted average of the trends from all other habitat types (deciduous woodland, mixed woodland, coniferous woodland, upland semi-natural open habitats, lowland semi-natural open habitats, arable farmland, pasture, mixed farming, wetlands and flowing water), accounting for their availability[31].

**Statistical modelling.** To account for temporal auto-correlation, all trend analyses (described above) included an AR(1) correlation structure. The AR(1) correlation structure was found to be optimal for time series modelling across the different response variables, based on the comparison of models with and without different autocorrelation structures (AR1 or AR2) using AIC and the examination of auto-correlation plots for the model residuals. There was no evidence of spatial auto-correlation in bird community indices across gardens within years, according to spline correlograms fitted to the raw data using the ncf R package[59]. When using GAM(M)s to investigate non-linear temporal trends, year was always fitted in the form of a thin-plate regression spine with a maximum of five degrees of freedom and the gamma parameter was fixed at 1.4 to reduce over-fitting. Generalised least squares (GLS, national-scale data) and linear mixed models (LMM, garden-scale data) were used to determine the significance of linear temporal trends when GAM (M)s fitted to the same data did not indicate non-linearity (e.g. the smoothed trend had one degree of freedom, was not significant, or did not deviate enough from the linear trend to be deemed ecologically meaningful). Significance was determined using maximum likelihood, Wald statistics, $\chi^2$ and F-tests as appropriate with alpha set at 0.05[60]. To identify periods of significant change within non-linear trends, where the rate of change (the slope) was distinguishable from zero given the uncertainty of the model, we estimated the first derivatives of the GAM temporal smooth[61,62]. A significant change was assumed where the 95% confidence intervals of the first derivatives excluded zero[61]. All analyses were performed using R version 3.4.3[63]. Trend analyses used the packages mgcv[64] and nlme[65], and phylogenetic comparative analyses used APE[66], phytools[67] and caper[68].

## Data availability

The Garden Bird Feeding Survey data and the bird feeding industry data that support the findings of this study are available upon reasonable request from the British Trust for Ornithology, https://www.bto.org/research-data-services. The UKCP09 temperature data used are available under licence from the British Met Office, https://www.metoffice.gov.uk/climate. The avian phylogeny data used are publicly available from BirdTree.org, https://www.BirdTree.org.

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

## Acknowledgements
We thank all the amateur ornithologists who have contributed to the collection of bird data; the past and present GBFS administrative team, particularly A. Prior, and C. Simm for overseeing the survey; K.W. Smith and L. Smith for discussions about and help with collecting the bird food industry data; D. Massimino for sharing the CBC/BBS bird trend annual indices; S. Gillings for help with examining bird spatial range shifts; J.W. Pearce-Higgins, S. Bearhop and K. Metcalfe for comments and suggestions on previous versions of the manuscript. This paper is the product of an institutional fellowship award (to K.E.P.), made possible thanks to a generous legacy donation to the BTO from Maxwell Hoggett.

## Author contributions
K.E.P. conceived the study and collated the bird food industry data. K.R and M.P.T. were responsible for the Garden Bird Feeding Survey data. K.E.P. conducted the analysis and drafted the paper, with substantial input from G.M.S. All authors contributed to the interpretation of results and manuscript completion.
