## [Peer Review File · Nature Communications]

Reviewer #1 (Remarks to the Author):

Overall comments

The manuscript “British bird communities have been shaped by long-term garden bird feeding” addresses an important topic, of high interest to the public, that has received very little attention from the scientific community – supplementary feeding of wild birds. This paper has three main facets: first, to attempt to quantify the changes to the bird feeding industry, and, thus, by proxy bird feeding practices, over a 40-year period by scoring bird-feeding-related advertising in a popular birding magazine; second, to determine how bird communities at feeders have changed over the same time-period; and third, to compare these changes with bird population patterns at a national scale. Bird feeding is occurring in many parts of the world on a phenomenal scale, and undoubtedly having an influence on bird communities, as the growing evidence in the literature shows. This study will make an important contribution to that literature and will be of interest to researchers in this field, given how little attention feeding has received in contrast to the magnitude of the practice.

Overall, I believe the paper highly relevant, with important findings and applications. I have concerns, however, regarding the primary conclusion made, and the lack of acknowledgement of the limitations of the data used and discussion of how bias in these data sets may influence patterns observed. Specifically, the conclusion that supplementary feeding has shaped bird communities in Britain is overstated, as it implies causation. The concluding paragraph is also very weakly constructed – overall, I feel that there is a lack of discussion of the findings in relation our current understanding and context of past studies. Furthermore, there the manuscript would benefit from improved clarity and conciseness in many places. These concerns are detailed below. Nevertheless, this study has the potential to make an important contribution to the literature, therefore, I urge the authors to improve the manuscript so that it is suitable for publication.

Main points

Ambitiously, this study sets out to determine whether bird feeding has influenced bird populations across Britain, which is no small task. Experimentally demonstrating such an influence across a scale this vast is logistically impossible, thus the authors have looked to existing long-term datasets to investigate patterns therein. This is virtually the only approach possible for answering questions of bird-feeding impact on a nationwide scale over multi-decadal timeframes. However, the main caveat of any study using this approach is that causation of any observed effects cannot be proven. The authors acknowledge this (L108–109), and yet still stretch their conclusions to suggest causation. The title itself implies the authors have found a causative effect of bird feeding on bird communities, and L14 states that the study reveals how community change at feeders “has translated into national population change” which certainly infers a causative effect. This is my primary criticism of

the paper – that, through the language they have chosen to use, the authors are inferring causation, when, at a maximum, they can only conclude that the evidence they have found supports the hypothesis that bird feeding has shaped bird communities. The conclusions need to be toned down throughout, using appropriate wording that does not allow for misinterpretation. Refer to Fuller et al. 2008 (citation #22) as a good example of appropriate interpretation of correlative data.

This paper claims novelty in that the authors suggest they are the first to look at bird feeding impact on bird communities, stating in the Summary Paragraph that the “impact on bird community ecology is unknown”. This is not true. They, in fact, cite a paper (#23, Galbraith et al. 2015) which experimentally tests and demonstrates impacts of feeding on bird communities. Yes, there have been few studies, and these may be from other geographic regions, but a blanket statement saying that the impacts are unknown is erroneous.

The study makes use of two existing datasets: the Garden Bird Feeding Survey (GBFS), and the Common Bird Census/Breeding Bird Survey. The authors interpret the findings under the assumption that the data from these sources are reliable and accurate. While this may be the case, a careful and considered interpretation should always examine bias. Here, there is very little scrutiny of how these datasets might be limited or biased. Two specific factors I would expect some mention of are how reliable citizen-science data are, and how the bird survey methodology may bias the observed results. Another assumption made is that advertising is a reliable index of changes in the feeding industry. This may be reasonable, but it is not explicitly justified in the paper. Are there other studies that have used advertising as a means of determining changes to an industry?

Given that the authors are concerned with bird communities, I am interested to know why community analysis methods (e.g. ordination, nMDS, PERMANOVA) were not employed. These non-parametric multivariate analyses give an understanding of whole communities, rather than reducing understanding of diversity down to a single numeric variable. These do have limitations, but they would be useful to supplement the current analyses. If these such methods were considered, but not used, why?

The methods are overly verbose and inaccessible in parts. They could do with streamlining, and reworking for clarity and conciseness.

Figure and Table captions contain extraneous information in places, and elsewhere do not contain enough information to stand alone.

Other comments

L15: what is meant by “systematic” increases in bird diversity? Clarify or remove.

L32–33: “Early feeding pioneers attracted a relatively simple bird community to gardens using kitchen scraps and table feeders.” No reference given for this statement.

L 56-58: “However, despite awareness that at least half of all British homeowners feed the birds in their gardens, it is yet to be established whether, and how, this has impacted upon wider bird population trends.” You should give appropriate reference to Fuller et al. 2008 (citation #22) here – this paper provides indicative evidence of bird feeding impacts on bird assemblages on a city scale. Because of what we know from Fuller et al. 2008, Galbraith et al. 2015, and the wider supplementary feeding literature, it is reasonable to predict larger-scale impacts on bird populations are occurring – this should be acknowledged as you establish the rationale for your study.

L61: “we uncovered 133 bird species using garden feeders during winter” – use identified, not uncovered.

L85–87: “It is reasonable to assume that increasing feeder numbers also reflect the greater variety in food types that became available during the same timeframe.” Why is this a reasonable assumption? If you make such an assumption you must justify it. I have certainly come across feeding practices which involve use of multiple feeders for the same food type.

L87–89: “Interestingly, when included as a covariate in the modeling of bird community temporal trends, the number of feeders was found to have a greater influence on species richness and diversity than either winter temperature or local habitat (Extended Data Fig. 4).” A point to consider here is feeder access. Interspecific interactions and hierarchies affect feeder access. By increasing the number of feeders, it allows greater opportunity for access, and therefore potentially reduces the ability for resource monopolization; i.e. more feeders enables more individuals to gain access at any one time, which may well enable more species to use feeders at any one time, regardless of food types provided.

L104–107: these two sentences essentially repeat the same information.

L111: “the relationship is unlikely to have resulted from birds ‘spilling over’ into gardens.” Be clear about what you mean here. Spilling over into gardens from where?

L111–114: “While many other factors are certain to influence interspecific variation in trends, these findings provide the first evidence that garden bird feeding is indeed influencing population growth

rather than simply sampling larger populations.” You do not explore any of the “other factors are certain to influence interspecific variation” – it is an omission not to discuss these, and demonstrate to the reader that you have considered what these might mean for your interpretation of the results. One major consideration is the different influences operating in urban areas. There is no mention of this throughout the paper. Also, as above, the results are not causative evidence. This is evidence that simply suggests there is influence. Tone down. Also, it is not clear what you mean by “rather than simply sampling larger populations.”

L115: “The fundamental basis for feeding wild birds is the perception that, by providing food during winter, one can improve the survival prospects of vulnerable individuals.” This is only one of the motivations for feeding birds – in a number of studies that investigate the motivations for bird feeding, a large proportion of feeding participants do so for enjoyment reasons.

L119: “We have presented unprecedented evidence” This is unnecessarily grandiose, particularly given that the conclusions overstep appropriate interpretation.

L130–137: This is a poor synthesis of the findings, and does not place them in the wider context of our current understanding. It is the first mention of the multiple mechanisms which might contribute to the observed patterns – this should be discussed in advance of a concluding paragraph. Equally, there are a number of potential negative feedback mechanisms, which are not mentioned at all in the paper, e.g. disease, competition. How do these fit with your findings?

L135: “That many species have increased feeder use across multiple generations (Fig. 4a) suggests ongoing natural selection” – This statement lacks critical, sound reasoning. Increased feeder use does not inherently suggest natural selection. Increased feeder use simply indicates more birds are using feeders which might be because populations have increased, or because more birds have discovered and are exploiting supplementary food (behavioural changes); it does not suggest these birds are being selected for because of their feeder use.

L135: “Individual decisions” – of people? Be specific.

L173: “a reliable estimate” – what do you mean by reliable? Be explicit.

L167: It is not explicit from the methods whether GBFS counts record birds only at/on feeders simultaneously, or if they can include feeder-visiting birds who are present in the vicinity. This is crucial for interpreting the results. As mentioned above, interspecific interactions and opportunities for feeder access determine who is using the feeder at any one time. The maximum number of birds using the feeder at any one time will not be the same as the number of birds in the vicinity of the feeder.

L189: “estimate of the total number of birds that feeders supported” – it is important to note that not all birds use feeders equally; there can be considerable variation in feeder use both within and between species (Crates, et al. 2016. Individual variation in winter supplementary food consumption and its consequences for reproduction in wild birds. *Journal of Avian Biology*; Jack, S.L. 2016. The Use of Supplementary Food Sources by Bird Communities and Individuals. Exeter, UK, University of Exeter; Galbraith, et al. 2017. Urban bird feeders dominated by a few species and individuals. *Frontiers in Ecology and Evolution* 5.)

L189: “it does provide a reliable means of comparing relative abundances across species.” Not necessarily. To the point above, the bird count method is crucial here. Depending on exactly which birds are included in the counts (at feeders only, or birds in the vicinity), the abundances recorded may or may not adequately reflect relative abundances. For instance, feeder access may allow for 10 birds total at one time. For a highly abundant species, they may take all 10 spots at the feeder, but there could be 30 more birds in a nearby tree waiting to gain access. For a less abundant species, they may take all 10 spots at the feeder, and no others may be waiting nearby.

L193: Why have you used the Simpson’s diversity index, over other diversity indices? Justify your method.

L209: “predominant surrounding local habitat” – be specific as to what is meant by this if possible.

L220: define “irregular” and “very low levels” of feeder use.

L220: “such that” – use “so that”.

L225: “such that” – use “so that”.

L229: “feeder use was fitted as the response term in the form of a binary trial against time period (start versus end of the time-series) using a generalised linear model” Clarity – do you mean feeder use was fitted as a binary response?

L244: “Wider countryside” – what do you mean by this? Define.

L259: “such that” – use “so that”.

L279: which parameters were fitted using Poisson vs. which were log-transformed?

L451-456: The figure caption contains extraneous information. The first sentence is too general – the figure only shows one index of change in the feeding industry. Generally you don't point out that dots on a figure are raw data points. Also y-axis title should be "Number of", not "Quantity of".

L466-475: This figure caption is verbose and contains extraneous information. The caption of a should simply state "The change in feeder use (proportion of gardens where species used feeders) between 1973 and 2012." You should not be explaining the figure legend in the text. The x-axis title for 3a does not correctly indicate what the axis represents – the axis title should simply be "Proportion of gardens". "Change in bird feeder usage" is what the data points on the figure show. Boxplots do not need an explanation in a figure caption.

L477: This table caption does not stand alone. It does not adequately explain what the table contains.

L481: "All species observed" – observed by whom? This is not stand alone.

L484: "Change in winter feeder use" – what is the specific parameter here? The table caption does not stand alone.

L486: "Level of significance is depicted using..." – denoted, not depicted.

L488-494: This figure does not show changes to the bird feeding industry, but changes to bird feeding related advertising over 33 years. This is an index of changes in the industry, but they are not the same thing. Placement of "(n= 33 years)" reads poorly at the end of this sentence.

L499-505: You should note why the axis is on such a scale. It is poor to refer to "averages" – do you mean the mean, median, or mode?

L507: "Significance of the fixed and smooth terms of the GAMM models used to examine the drivers of variation in Simpson's diversity and species richness of birds using feeders in gardens" – unnecessary to point out that this table shows significance of the model results. Saying "Results of GAMM models testing variation in..." is sufficient.

General comment on all figures: it is extraneous to be using “trend in” to describe figures. The figure caption should not spell out that a trend is or is not present – this interpretation should be restricted to results and discussion.

Reviewer #2 (Remarks to the Author):

I enjoyed reading your manuscript, and while you will notice that I have made a substantial number of comments below, the spirit in which I intend them is as indications or suggestions of ways in which I think that you can make the story being told in your current manuscript more robust and convincing for readers. There are two general themes to my comments. First, given that your data, as it typical for citizen science data, were not collected for the specific purpose of your study, and as such data analyses and interpretations need to consider sources of random noise and systematic bias that potentially are alternative explanations for patterns that emerge from analyses. I thought that your current manuscript was focused so tightly on your proposed explanation that potential alternative, or at least more nuanced, interpretations were ignored. Second, I expect that you have given this first issue serious consideration already, but there were a number of locations through the manuscript where I suggest that you should provide readers with just little bits of additional information, for example mention of why certain decisions were made regarding the analyses that you performed, and making currently implicit assumptions explicit in the text of the paper. I think that the overall goal of any revisions is to make it easy for readers to see (without making them think too much!) that you have done everything possible to have produced non-spurious patterns from your statistical analyses, and that you have explored and dismissed potential alternative explanations for the patterns that you have found. Having written this, I will note that I think that your analyses and interpretations are all reasonable; what I think needed is a bit more work to make your case completely convincing. My comments vary from suggestions of items about which you could think through (and then potentially dismiss as irrelevant) to items that I feel strongly do need to be addressed either through making changes to the text to increase clarity or possibly even to conducting additional analyses to confirm that your observed patterns are biologically real. I hope that my wording will indicate to you where along this gradient each comment exists.

Each of the comments below is connected to a specific line number or range of line numbers in the PDF version of your manuscript that I was given to review. My comments are as follows:

Line 74: "numerical dominance among species" is a phrase whose meaning is not immediately clear. Perhaps rephrase or expand; otherwise readers could mentally stumble when they encounter this phrase and wonder why the two patterns noted at the start of the sentence "suggest" anything.

Line 76: What does "relative abundance" mean within the context of this sentence? I think that this should be defined in the main text. I do not think that you are meaning the sums of the maximum

counts of the 5 most common species, but instead values from Fig. 2e; the y-axis of this figure panel is one type of relative abundance (i.e. as a proportion of the total number of birds summed across all species), but there are other interpretations of what "relative abundance" means, including just the reported counts (where "relative" indicates that the counts are assumed to be some index of actual abundance). I have seen this latter meaning used frequently in bird monitoring literature.

Lines 87-89: Shouldn't the information in this last sentence be presented as the second last sentence, and the current second last sentence become last in this paragraph? I.e. the information in the last sentence is a pattern whose explanation/interpretation is given in the current second last sentence?

Lines 92-93: Possible alternative explanation: Spatial range expansion is bringing many wintering species in contact with a greater proportion of all feeder sites?

More generally, is there any geographic variation in the observed patterns. The current analyses are implicitly assuming that Great Britain can be treated as a uniform area.

Line 99: Placed where it is, this parenthetical information from the phylogenetic regression could be interpreted as meaning that the statistics are describing the test of the phylogenetic effect...which I presume they are not. What about shifting the parenthetical test to immediately after "timeframe", a bit earlier in this sentence, to bring the statistics in direct contact with the statement of pattern that the statistics are supporting?

Lines 102-103: Are these the regular/habitual feeder users? Or are these whatever assortment of 72 species for which population trends were available, some of which are and are not regular feeder users. Remind readers about this somewhere here.

Lines 109-111: I think that you've done yourselves a dis-service by adding this information almost as an afterthought to the analysis of population trends in human-dominated habitats. A comparison of population trends in human-dominated and "non-human landscapes" (that's a sort of awkward term, perhaps replace with "non-anthropogenic"...although that's not exactly right either in reference to pretty much anywhere in Great Britain) is the strongest evidence that you could produce that bird feeding is directly impacting bird species' population sizes.

I suggest that you frame this entire paragraph's contents as a formal contrast between population trends in human-dominated and non-human-dominated landscapes, and reanalyse your data in a way that would allow the human/non-human landscape contrast to be formally tested.

Of course, statistical significance in any such test would imply that your overall conclusion would have to be nuanced a little bit, such that humans are having a dramatic effect on Great Britain's bird communities in places where humans are feeding birds (and not everywhere, as the current conclusion is suggesting).

Line 172: Does this mean that migratory as well as resident species are included, given that 20 or 26 weeks would create a span of time from late fall migration to early spring migration? I think that readers should be told this in the text of the paper.

Line 181: This data-filtering decision also allows for changes in migration dates of bird species through time to affect the measured changes through time in community structure. Is this a problem/source of bias?

Line 185-186: Is the maximum affected by not just how many weeks had data at a site, but which weeks?

The implicit assumption of saturation/asymptoting counts with the 20 weeks of effort seems reasonable to me, but it might still be good to mention (and maybe for a few species test) that this assumption is valid.

Lines 188-189: Hmm...as written I do not think that this statement is true. You are writing that one can directly compare the relative abundance values from species to species, but this assumes that an equal proportion of individuals within each species around a feeder site was not counted of each species. This assumption of equal detectability, and its recognized falsehood, is what motivated development of survey methods like occupancy modelling and distance sampling, so the inappropriateness of the assumption is widely recognized.

It does seem a lot more reasonable to me that one could compare the changes, from beginning to end of the study period, in relative abundance values across species, which is what I presume you actually want to do.

Line 191: This isn't a true statement: you calculated 3 indices. Figure 2e shows an evenness statistic.

Phrased another way, should the "Bird community indices" and "National-scale trends in bird communities" text be in closer proximity, and at the current location of the latter sub-section of the Methods?

Line 195: Would it be useful to mention that a comparison of richness and Simpson's index values is informative, e.g., if richness is unchanged and diversity increases, then one can infer that evenness increases?

Line 202: Given that you used bootstrapping, you should also have generated estimates of confidence for each year. Did statistical confidence vary systematically from start to end of the survey (such that any assumption of equal variance through time for regression purposes would be inappropriate), or was the variation around estimated means asymmetrical (such that calculating and interpreting means would be inappropriate)?

Line 208: My own experience is that participant-based classification along a rural to urban gradient is inaccurate across larger geographic areas, because participants' perceptions of what constitutes "rural" will vary depending on where they live. Specifically, I have seen that people living near larger cities will consider areas as being rural, whereas someone in a less populated area would not classify these same sites as being rural. I am basing this on having compared participant classification with land cover information from remotely sensed sources.

Line 212: I think that the reason for your choice to use the average should be explained, at least in a sentence. I'm suggesting this because my immediate reaction upon reading this text is that bird communities or feeder use may not be influenced as much by average temperature as by more extreme weather events.

Lines 226-228: Why did you choose to use this approach, and not fit a regression through all of the data? I am not suggesting that your approach is wrong, but I (and I think other readers) would like to know the reason for comparing beginning to end, without examining whether there has been a monotonic change through time. This assumption of monotonicity is implicit in the method that you have chosen, is it not?

Given that new food types were introduced at different points in time, we might expect non-linear increases in feeder use through time to be seen. I am not suggesting that analyses such as these should form part of the material presented in the paper. However, I am wondering whether you have looked at the pattern of change in feeder use through time (e.g., fitting a species' data with a GAM to visualize the shape of the change in abundance), which would allow you to state in the manuscript that you have verified your implicit assumption that the changes in feeder use that you observed were monotonic.

Lines 229-231: I think that a bit more information would be useful, or at least interesting, for readers here. First, I presume that converting the raw data, which are in the form of maximum counts of birds at each in each year, into presence/absence data and thus lowering the information content of the data was deliberate. However, the reason is not explained to readers. Second, for the sake of clarity I think that the text should be altered slightly to make clearer that each site was represented up to 3 times in the "start" and 3 times in the "end" data. I presume that this was the case, but for the sake of allowing readers to fully understand and potentially replicate the analyses, your actual treatment of the data should be described just a bit more clearly.

Lines 217-233: Both with the calculation of the change in feeder usage and the change in population index, you are using statistical modelling to create an estimate of something, and each of these estimates has an error associated with it. Then differences between estimates from the "start" and "end" periods are calculated, but the uncertainties from the original estimates are discarded. Finally, these two differences are regressed against each other and a p-value is calculated that ignores the uncertainty that was not carried through from the initial estimates. Minimally, I believe that this means that the statistical probability of association in the final regression is too small by some unknown amount, because sources of error were discarded in the processing pipeline. I think that minimally this deserved to be at least recognized in the paper's text. Additionally, is there any

possible way that not propagating errors through the processing could cause the pattern that you observed as an artifact? I cannot see how this could happen, I will add, but I think that it is a question that you should consider.

Lines 259-260: I'm not entirely sure, but I think that the refitting would not serve to allow statistical significance to be "better approximated", particularly if the GAMM "shrank" the smoothing trend to be a linear regression. For near-linear trends though, simplifying to a straight line would have the advantage that you would have a simple slope parameter that is easy to interpret and is automatically produced as output.

Lines 261-263: Where the spline correlograms fitted to the residuals from GAMMs and LMMs? I presume that this is the case, but you need to state which type of information was explored for spatial autocorrelation.

Lines 280-283: You have explained what you have done (i.e. controlling for changes in magazine length, and the amount of advertising through time), but you have not explained why these corrections are necessary. I.e., there are implicit assumptions that I think should be stated.

To me, it looks like you are assuming that fewer types of bird food will be advertised if there is less space in which to advertise...but wouldn't it make sense for a company marketing a product to deliberately differentiate themselves by advertising a novel food item (i.e. the number of unique food items should be relatively independent of available space, at least beyond some amount of available space...suggesting non-linear relationships between available space and the number of unique food types)? So, could your statistical corrections possibly be resulting in an underestimate of the change through time if you are forcing linear corrections?

Line 290: You described how there was a saturation in species richness at sites after there were at least 20 weeks of observations submitted for a site in a season. However, does the abundance (i.e. the maximum number of birds observed at a single time) also saturate at this point, or does abundance increase with the number of observation periods, beyond the minimal 20 observation periods, for any species?

I would be surprised if this was an important effect, but possibly something worth checking and mentioning in passing at the same time that the saturation in richness is mentioned on line 172?

Lines 295-296: Why do you need to write "using the LMM modelling framework" when "using LMMs" says the same thing?

Lines 303-304: Would direct comparability after standardizing be possible if the values of the predictor variables were not normally (or at least symmetrically) distributed around their mean? So, are you sure that the interpretation based on direct comparability is legitimate for your data, or

could interpretations have been biased? I do not know the answer to my question, but this was a question that came to mind as I read this text.

Figure 3, panel A: So, increased feeder use means a greater diversity of food is being provided, thus attracting a greater diversity of bird species. At least, that's the presumption. Presumably then it should be the "non-traditional" feeder birds that benefitted most (and increased in population size the most). However, wouldn't that mean that the PGLS should show an effect of phylogeny because these non-traditional feeder birds should be more taxonomically similar? I.e., is there some biological meaning that can be obtained from the lack of a phylogenetic signal, and if so would it be worth mentioning in the paper?

Figure 3, panel A: Strictly speaking the wording on the x-axis is wrong, I think. The values plotted on the axis are not changes, but actual proportions of sites with species attending feeders in each of two years.

Extended Data Table 3: I think that it would be useful to anyone scanning this table if the species that you define as feeder users were indicated (e.g., by putting their names in bold font). Otherwise, readers will have to cross-reference with another location to find this information.

Reviewers' comments

Reviewer #1:

Overall comments

The manuscript “British bird communities have been shaped by long-term garden bird feeding” addresses an important topic, of high interest to the public, that has received very little attention from the scientific community – supplementary feeding of wild birds. This paper has three main facets: first, to attempt to quantify the changes to the bird feeding industry, and, thus, by proxy bird feeding practices, over a 40-year period by scoring bird-feeding-related advertising in a popular birding magazine; second, to determine how bird communities at feeders have changed over the same time-period; and third, to compare these changes with bird population patterns at a national scale. Bird feeding is occurring in many parts of the world on a phenomenal scale, and undoubtedly having an influence on bird communities, as the growing evidence in the literature shows. This study will make an important contribution to that literature and will be of interest to researchers in this field, given how little attention feeding has received in contrast to the magnitude of the practice.

Overall, I believe the paper highly relevant, with important findings and applications. I have concerns, however, regarding the primary conclusion made, and the lack of acknowledgement of the limitations of the data used and discussion of how bias in these data sets may influence patterns observed. Specifically, the conclusion that supplementary feeding has shaped bird communities in Britain is overstated, as it implies causation. The concluding paragraph is also very weakly constructed – overall, I feel that there is a lack of discussion of the findings in relation our current understanding and context of past studies. Furthermore, there the manuscript would benefit from improved clarity and conciseness in many places. These concerns are detailed below. Nevertheless, this study has the potential to make an important contribution to the literature, therefore, I urge the authors to improve the manuscript so that it is suitable for publication.

- We would like to thank the reviewer for their kind words about our study and its important contribution to the literature.
- We have made consideration changes to the manuscript to address the reviewer’s concerns. In particular we have added more explicit detail about the data used and we have included more in-depth interpretation of our findings in order to support the conclusions made.

Main points

Ambitiously, this study sets out to determine whether bird feeding has influenced bird populations across Britain, which is no small task. Experimentally demonstrating such an influence across a scale this vast is logistically impossible, thus the authors have looked to existing long-term datasets to investigate patterns therein. This is virtually the only approach possible for answering questions of bird-feeding impact on a nationwide scale over multi-decadal timeframes. However, the main caveat of any study using this approach is that causation of any observed effects cannot be proven. The authors acknowledge this (L108–109), and yet still stretch their conclusions to suggest causation. The title itself implies the authors have found a causative effect of bird feeding on bird communities, and L14 states that the study reveals how community change at feeders “has translated into national

population change” which certainly infers a causative effect. This is my primary criticism of the paper – that, through the language they have chosen to use, the authors are inferring causation, when, at a maximum, they can only conclude that the evidence they have found supports the hypothesis that bird feeding has shaped bird communities. The conclusions need to be toned down throughout, using appropriate wording that does not allow for misinterpretation. Refer to Fuller et al. 2008 (citation #22) as a good example of appropriate interpretation of correlative data.

- We agree, this study cannot definitively infer causation – as the reviewer acknowledges, this is currently impossible. Indeed, as the reviewer also notes, we had already explicitly acknowledged that we do not show causation within the manuscript. However, we have used the best available resources and analytical approaches to test the consequences of large-scale, long-term garden bird feeding in Britain. As such, we believe the conclusions being drawn are fully supported, particularly in light of the additional study rationale, data exploration and results interpretation now included following helpful suggestions from the reviewers.
- However, we acknowledge the point raised and certainly do not intend for the conclusions to be misinterpreted, and so we have rephrased the text throughout to avoid any undue suggestion of proof of causation. For example, in the **Abstract**, the text has been changed from “...show how this has translated into national population change” to “...provide evidence to suggest how this has led to bird population changes...” (L10-12). In the **Results and Discussion**, “...evidence that garden bird feeding is indeed influencing population growth...” has been changed to “...landscape scale evidence consistent with garden bird feeding having influenced population growth...” (L178-180). And “...unprecedented evidence that this habitual human activity is driving the national-scale restructuring of bird communities” has been changed to “unique evidence that this habitual human activity is associated with the national-scale restructuring of bird communities” (L188-189).

This paper claims novelty in that the authors suggest they are the first to look at bird feeding impact on bird communities, stating in the Summary Paragraph that the “impact on bird community ecology is unknown”. This is not true. They, in fact, cite a paper (#23, Galbraith et al. 2015) which experimentally tests and demonstrates impacts of feeding on bird communities. Yes, there have been few studies, and these may be from other geographic regions, but a blanket statement saying that the impacts are unknown is erroneous.

- Our study goes significantly further than previous research on the community impacts of garden bird feeding, not least in terms of spatio-temporal scale (no previous study has considered such a large scale), and therefore its novelty is indisputable.
- To avoid any confusion, we have made it much clearer, in the **Abstract (L9-10)** and throughout (e.g. **L39-41, L210-212**), that we are referring to large-scale, long-term community patterns.

The study makes use of two existing datasets: the Garden Bird Feeding Survey (GBFS), and the Common Bird Census/Breeding Bird Survey. The authors interpret the findings under the assumption that the data from these sources are reliable and accurate. While this may be the case, a careful and considered interpretation should always examine bias. Here, there is very little scrutiny of how these datasets might be limited or biased. Two specific factors I would expect some mention of are how reliable citizen-science data are, and how the bird survey methodology may bias the observed

results. Another assumption made is that advertising is a reliable index of changes in the feeding industry. This may be reasonable, but it is not explicitly justified in the paper. Are there other studies that have used advertising as a means of determining changes to an industry?

Regarding citizen science data limitations and bias

- We do not have space and it would be beyond the scope of this paper to provide a detailed assessment of the value and limitations of citizen science data, not least because this covers a wide range of types of data collection. However, both of the schemes considered here involve structured sampling to a defined protocol, in which effort is measurable and/or standardized, with individuals sampling for multiple years. Analytical techniques and data interpretation are also well-established. For example, CBC/BBS trends inform national biodiversity indicators in the UK and both datasets have also underpinned many peer-reviewed papers over several decades.
- There is no reason to suspect specific bias due to the fact that these surveys are conducted by volunteers. Structured sampling controls data collection methods, site selection in the BBS is random and CBC plot locations have been shown to be representative of land-use in lowland Britain (Fuller et al. 1985). GBFS site locations are managed by the survey organiser, focusing on suburban/urban and rural garden habitats and providing national coverage (see Supplementary Figure 2). Volunteer field skills are not formally controlled, but there is no reason to suspect spatial biases or temporal change in patterns of, for example, count accuracy or identification skill. Hence, any areas where observer skill falls below what might be expected of professional surveyors (who are also never error-free) should be causes of noise, not bias. In addition, all data input and processing systems incorporate checks for extreme values and unexpected species, so trapping field and inputting errors.
- We now explicitly address the reliability of the GBFS data in the **Methods (L260-261 and L275-279)**.
- We have also added a statement summarising the CBC and BBS survey design and data value (**L361-363**).

Regarding using advertising as an index of garden feeding industry change

- We have added explicit justification for using advertising data as an index of change in the bird feeding industry in the main text (**L57-63**). Notably, marketing theory recognises the close association between advertising and consumer demand and has demonstrated the impacts of brand advertising on total industry demand. By virtue of this, advertising of garden bird feeding products should be indicative of consumers' garden bird feeding habits.

Given that the authors are concerned with bird communities, I am interested to know why community analysis methods (e.g. ordination, nMDS, PERMANOVA) were not employed. These non-parametric multivariate analyses give an understanding of whole communities, rather than reducing understanding of diversity down to a single numeric variable. These do have limitations, but they would be useful to supplement the current analyses. If these such methods were considered, but not used, why?

- Considerable thought went into the community metrics and analytical methods used, given the complexity of quantifying, analysing and presenting patterns of temporal variation in

community composition. Our approach was to use a combination of analytical techniques that would clearly convey broad patterns of bird community change, while also being accessible to a diverse audience interested in anthropogenic impacts on wildlife.

- Although, as the reviewer mentions, species richness and Simpson's diversity measures reduce the community to a single numeric variable, so to speak, the analyses we have performed do give an understanding of changes across the whole community. Not least, *k*-dominance curves provide evidence of the abundance distribution of all species within the community, allowing the differences in community structure over time to be clearly demonstrated. Also, by comparing national- and garden-scale trends in species richness and species diversity this has enabled us to unpick the underlying species patterns that have contributed to community change more broadly. These analyses have been supplemented by a more detailed investigation of species-level changes over time, providing further evidence of the variability within the community and its wider implications.
- The non-parametric multivariate analyses mentioned by the reviewer are essentially an alternative way of showing the same results/patterns already described. However, the primary reason for not using them is that they would be complicated to present and interpret for multiple time points. In light of this, and given that the reviewer does not find fault with the analyses we have used, we do not see a compelling reason to use any of the alternative community analysis methods mentioned.

The methods are overly verbose and inaccessible in parts. They could do with streamlining, and reworking for clarity and conciseness.

- In wanting to make sure the Methods were thorough and clear, we acknowledge that they were verbose. **The Methods have been substantially revised**, and we have made every effort to streamline them by removing superfluous detail in order to improve clarity and conciseness, while also needing to include additional information to address other comments by the reviewers.

Figure and Table captions contain extraneous information in places, and elsewhere do not contain enough information to stand alone.

- All Figure and Table captions have been edited to maintain consistency, while also meeting the specific guidelines for *Nature Communications*.

Other comments

L15: what is meant by “systematic” increases in bird diversity? Clarify or remove.

- ‘Systematic’ has been removed.

L32–33: “Early feeding pioneers attracted a relatively simple bird community to gardens using kitchen scraps and table feeders.” No reference given for this statement.

- Reference added.

L 56-58: “However, despite awareness that at least half of all British homeowners feed the birds in their gardens, it is yet to be established whether, and how, this has impacted upon wider bird

population trends.” You should give appropriate reference to Fuller et al. 2008 (citation #22) here – this paper provides indicative evidence of bird feeding impacts on bird assemblages on a city scale. Because of what we know from Fuller et al. 2008, Galbraith et al. 2015, and the wider supplementary feeding literature, it is reasonable to predict larger-scale impacts on bird populations are occurring – this should be acknowledged as you establish the rationale for your study.

- Thank you for this suggestion. Contextual details from Fuller et al 2008, Galbraith et al 2015 and others have been added to the **Introduction (L33-47)**, helping us to establish the rationale for our study more effectively.

L61: “we uncovered 133 bird species using garden feeders during winter” – use identified, not uncovered.

- Corrected.

L85–87: “It is reasonable to assume that increasing feeder numbers also reflect the greater variety in food types that became available during the same timeframe.” Why is this a reasonable assumption? If you make such an assumption you must justify it. I have certainly come across feeding practices which involve use of multiple feeders for the same food type.

- This assumption is reasonable because, given the psychology behind bird feeding, people are more likely to buy multiple feeders to diversify types of food provision and to attract more species, rather than to attract more of the same species. Of course, there will be exceptions to this, but it is likely to be the general pattern, especially given the growing diversity of foods and feeders, which offers an additional reason for adding a new feeder.
- Justification of this assumption has been added to the text (**L105-108**).

L87–89: “Interestingly, when included as a covariate in the modeling of bird community temporal trends, the number of feeders was found to have a greater influence on species richness and diversity than either winter temperature or local habitat (Extended Data Fig. 4).” A point to consider here is feeder access. Interspecific interactions and hierarchies affect feeder access. By increasing the number of feeders, it allows greater opportunity for access, and therefore potentially reduces the ability for resource monopolization; i.e. more feeders enables more individuals to gain access at any one time, which may well enable more species to use feeders at any one time, regardless of food types provided.

- Thank you for this suggestion. We have added considerably more detail to help with the interpretation of this result, including discussion of interspecific dominance hierarchies and resource monopolisation as suggested (**L119-126**).

L104–107: these two sentences essentially repeat the same information.

- Rephrased and repetition removed.

L111: “the relationship is unlikely to have resulted from birds ‘spilling over’ into gardens.” Be clear about what you mean here. Spilling over into gardens from where?

- Rephrased for clarity (**L218-222**).

L111–114: “While many other factors are certain to influence interspecific variation in trends, these findings provide the first evidence that garden bird feeding is indeed influencing population growth

rather than simply sampling larger populations.” You do not explore any of the “other factors are certain to influence interspecific variation” – it is an omission not to discuss these, and demonstrate to the reader that you have considered what these might mean for your interpretation of the results. One major consideration is the different influences operating in urban areas. There is no mention of this throughout the paper. Also, as above, the results are not causative evidence. This is evidence that simply suggests there is influence. Tone down. Also, it is not clear what you mean by “rather than simply sampling larger populations.”

- The sentence has been rephrased to tone it down (L178-180).
- Our aim here is to make it clear that we are not claiming that feeding is the only, or necessarily the most dominant influence, just that it has an effect. We are purposely concise, as we do not think that this paper is the place for a comprehensive discussion of causes of population change in terrestrial birds. However, we have outlined some examples of ‘other factors’ to the text to acknowledge their potential influence (L129-132).
- Regarding ‘the different influences in urban areas’ – We show evidence consistent with an effect of feeding, independent of any of urban land-use or directly associated factors (**now more clearly stated L159-162**). Again, we are not trying to produce a comprehensive assessment of the drivers of urban bird population change.
- To clarify, “*rather than simply sampling larger populations*” means that the patterns within gardens are not simply a reflection of wider population changes, with garden feeders functioning as a kind of assay cf. a pitfall trap. We have included further interpretive detail to make this point more clearly (L211-225).

L115: “The fundamental basis for feeding wild birds is the perception that, by providing food during winter, one can improve the survival prospects of vulnerable individuals.” This is only one of the motivations for feeding birds – in a number of studies that investigate the motivations for bird feeding, a large proportion of feeding participants do so for enjoyment reasons.

- We have rephrased this sentence to make it clear that we are referring to the historical origins of wild bird feeding and acknowledged the well-being benefits association with feeding (L181-185).

L119: “We have presented unprecedented evidence” This is unnecessarily grandiose, particularly given that the conclusions overstep appropriate interpretation.

- “Unprecedented” has been replaced with “unique”.

L130–137: This is a poor synthesis of the findings, and does not place them in the wider context of our current understanding. It is the first mention of the multiple mechanisms which might contribute to the observed patterns – this should be discussed in advance of a concluding paragraph. Equally, there are a number of potential negative feedback mechanisms, which are not mentioned at all in the paper, e.g. disease, competition. How do these fit with your findings?

- A description of the potential mechanisms by which feeder use could influence bird populations has been incorporated into the **Introduction**, including acknowledging the existence of both positive and negative impacts (L33-41). We have also expanded our

interpretation of the findings throughout the **Results and Discussion** to better place them in the wider context of our current understanding.

- We have also added more detail about mechanisms to the final paragraph of the **Discussion (L201-210)**, including specific reference to the known negative impacts of supplementary feeding.

L135: “That many species have increased feeder use across multiple generations (Fig. 4a) suggests ongoing natural selection” – This statement lacks critical, sound reasoning. Increased feeder use does not inherently suggest natural selection. Increased feeder use simply indicates more birds are using feeders which might be because populations have increased, or because more birds have discovered and are exploiting supplementary food (behavioural changes); it does not suggest these birds are being selected for because of their feeder use.

- This statement has been removed.

L135: “Individual decisions” – of people? Be specific.

- ‘...by homeowners’ added for clarity.

L173: “a reliable estimate” – what do you mean by reliable? Be explicit.

- “Reliable” has been replaced with “accurate (asymptotic)”.

L167: It is not explicit from the methods whether GBFS counts record birds only at/on feeders simultaneously, or if they can include feeder-visiting birds who are present in the vicinity. This is crucial for interpreting the results. As mentioned above, interspecific interactions and opportunities for feeder access determine who is using the feeder at any one time. The maximum number of birds using the feeder at any one time will not be the same as the number of birds in the vicinity of the feeder.

- The instructions given to GBFS participants can be viewed online at <https://www.bto.org/volunteer-surveys/gbfs/taking-part/instructions>. These are easily accessed from the survey website link provided in the Methods. The specific instructions regarding bird counts are as follows: “*Please enter the highest number (peak count) of birds of each species observed feeding or drinking at any one time (i.e. simultaneously) during the week. Record only those birds you observe feeding on food-stuffs or drinking water which has been deliberately put out for them within your defined feeding area (i.e. do not include natural fruits growing on shrubs and trees)*”.
- The intention is that participants *record the number of birds seen simultaneously, using the feeders* (i.e. on and in the vicinity). For example, if a flock of 10 House Sparrows was on and around a feeder, but only 5 were physically sitting on the feeder at any one time, participants would record 10.
- This is now stated explicitly in the **Methods (L249-251)**.

L189: “estimate of the total number of birds that feeders supported” – it is important to note that not all birds use feeders equally; there can be considerable variation in feeder use both within and between species (Crates, et al. 2016. Individual variation in winter supplementary food consumption and its consequences for reproduction in wild birds. *Journal of Avian Biology*; Jack, S.L. 2016. The

Use of Supplementary Food Sources by Bird Communities and Individuals. Exeter, UK, University of Exeter; Galbraith, et al. 2017. Urban bird feeders dominated by a few species and individuals. *Frontiers in Ecology and Evolution* 5.)

- For clarity, this has been rephrased as “...*estimate of the total number of birds visiting feeders...*”.

L189: “it does provide a reliable means of comparing relative abundances across species.” Not necessarily. To the point above, the bird count method is crucial here. Depending on exactly which birds are included in the counts (at feeders only, or birds in the vicinity), the abundances recorded may or may not adequately reflect relative abundances. For instance, feeder access may allow for 10 birds total at one time. For a highly abundant species, they may take all 10 spots at the feeder, but there could be 30 more birds in a nearby tree waiting to gain access. For a less abundant species, they may take all 10 spots at the feeder, and no others may be waiting nearby.

- This statement is supported by the bird count methodology used (now stated explicitly in the **Methods L249-251**), as described in our earlier response.

L193: Why have you used the Simpson’s diversity index, over other diversity indices? Justify your method.

- Simpson’s diversity index was used based on its popularity and the recommendation of Magurran (2004) *Measuring biological diversity*, where it is described as one of the most robust and meaningful measures of diversity. It also complements species richness. We have made this clearer in the **Methods (L315-318)**.

L209: “predominant surrounding local habitat” – be specific as to what is meant by this if possible.

- The survey instructions give the following guidance: ‘Urban’ means *densely built-up areas and town centres with very few natural or near-natural bird feeding sites*. ‘Suburban’ means *inhabited areas near countryside or with large gardens, municipal parks or recreational areas*. ‘Rural’ refers to *areas away from towns, with just a few scattered houses, farms or other isolated buildings*. This has been summarised in the **Methods (L297-300)**.
- Please also see the response to Reviewer #2’s L208 comment.

L220: define “irregular” and “very low levels” of feeder use.

- These terms have been removed from the revised Methods.

L220: “such that” – use “so that”.

- “such that” has been removed throughout.

L225: “such that” – use “so that”.

- “such that” has been removed throughout.

L229: “feeder use was fitted as the response term in the form of a binary trial against time period (start versus end of the time-series) using a generalised linear model” Clarity – do you mean feeder use was fitted as a binary response?

- The description of this analysis has been edited for clarity (**L353-355**).

L244: “Wider countryside” – what do you mean by this? Define.

- We have changed this to ‘non-urban’ for clarity (see **description in L390-396**).

L259: “such that” – use “so that”.

- “such that” has been removed throughout.

L279: which parameters were fitted using Poisson vs. which were log-transformed?

- The structure of the models applied to each bird feeding industry response are now stated explicitly in the revised **Methods (L227-240)**.

L451-456: The figure caption contains extraneous information. The first sentence is too general – the figure only shows one index of change in the feeding industry. Generally you don’t point out that dots on a figure are raw data points. Also y-axis title should be “Number of”, not “Quantity of”.

- The legend for **Figure 1** has been shortened to remove extraneous information.
- The figure shows temporal changes in the quantity of food available *and* evidence of temporal patterns of food diversification. Therefore the title (first sentence) has not been changed, since it is both accurate and meets the journal guidelines.
- The y-axis has been changed as advised.

L466-475: This figure caption is verbose and contains extraneous information. The caption of a should simply state “The change in feeder use (proportion of gardens where species used feeders) between 1973 and 2012.” You should not be explaining the figure legend in the text. The x-axis title for 3a does not correctly indicate what the axis represents – the axis title should simply be “Proportion of gardens”. “Change in bird feeder usage” is what the data points on the figure show. Boxplots do not need an explanation in a figure caption.

- The caption and x-axis title for 3a have been changed as advised, and we have made edits to reduce the total length of the legend for **Figure 3**.
- The explanation of the boxplot is included on the recommendation of Krzywinski, M. & Altman, N. (2014) Visualizing samples with box plots. *Nature Methods*, **11**, 119. Specifically, they state “*Aspects of the box plot such as width, whisker position, notch size and outlier display are subject to tuning; it is therefore important to clearly label how your box plot was constructed. Fewer than 20% of box plot figures in 2013 Nature Methods papers specified both sample size and whisker type in their legends—we encourage authors to be more specific*”. On this basis, we have not removed the boxplot description but we are happy for the journal’s editorial team to advise further.

L477: This table caption does not stand alone. It does not adequately explain what the table contains.

- This table has been removed and the results incorporated into the main text.

L481: “All species observed” – observed by whom? This is not stand alone.

- “within GBFS” added to the title for **Supplementary Table 2** for clarity

L484: “Change in winter feeder use” – what is the specific parameter here? The table caption does not stand alone.

- The title for **Supplementary Table 3** has been expanded and the definition of feeder use (the response parameter) is now reiterated in the table legend.

L486: “Level of significance is depicted using...” – denoted, not depicted.

- Corrected

L488-494: This figure does not show changes to the bird feeding industry, but changes to bird feeding related advertising over 33 years. This is an index of changes in the industry, but they are not the same thing. Placement of “(n= 33 years)” reads poorly at the end of this sentence.

- The figure legend for **Supplementary Figure 1** conforms to the journal guidelines, which ask for a ‘brief title’. The phrasing of the title matches that used in the main text and Methods for ease of reference. Therefore we believe it would be detrimental to alter it, but we are happy for the journal’s editorial team to advise further. We note that exact details of the data and indices used are provided in the description immediately following the title.
- The sample size has been moved.

L499-505: You should note why the axis is on such a scale. It is poor to refer to “averages” – do you mean the mean, median, or mode?

- **Supplementary Figure 3:** Details of y-axis scaling have been added to the legend and ‘mean’ is now used in place of ‘on average’.

L507: “Significance of the fixed and smooth terms of the GAMM models used to examine the drivers of variation in Simpson’s diversity and species richness of birds using feeders in gardens” – unnecessary to point out that this table shows significance of the model results. Saying “Results of GAMM models testing variation in...” is sufficient.

- **Supplementary Figure 4:** Revised as directed.

General comment on all figures: it is extraneous to be using “trend in” to describe figures. The figure caption should not spell out that a trend is or is not present – this interpretation should be restricted to results and discussion.

- ‘Trend’ is conventional terminology used to describe patterns of long-term change in time-series analyses. We have used it on this basis throughout the manuscript. It is neutral and not an interpretation of the temporal patterns, and therefore we do not believe it is necessary to remove “trend in” from figure captions. However, we are happy for the journal’s editorial team to advise further.

Reviewer #2:

Overall comments

I enjoyed reading your manuscript, and while you will notice that I have made a substantial number of comments below, the spirit in which I intend them is as indications or suggestions of ways in which I

think that you can make the story being told in your current manuscript more robust and convincing for readers. There are two general themes to my comments. First, given that your data, as it typical for citizen science data, were not collected for the specific purpose of your study, and as such data analyses and interpretations need to consider sources of random noise and systematic bias that potentially are alternative explanations for patterns that emerge from analyses. I thought that your current manuscript was focused so tightly on your proposed explanation that potential alternative, or at least more nuanced, interpretations were ignored. Second, I expect that you have given this first issue serious consideration already, but there were a number of locations through the manuscript where I suggest that you should provide readers with just little bits of additional information, for example mention of why certain decisions were made regarding the analyses that you performed, and making currently implicit assumptions explicit in the text of the paper. I think that the overall goal of any revisions is to make it easy for readers to see (without making them think too much!) that you have done everything possible to have produced non-spurious patterns from your statistical analyses, and that you have explored and dismissed potential alternative explanations for the patterns that you have found. Having written this, I will note that I think that your analyses and interpretations are all reasonable; what I think needed is a bit more work to make your case completely convincing. My comments vary from suggestions of items about which you could think through (and then potentially dismiss as irrelevant) to items that I feel strongly do need to be addressed either through making changes to the text to increase clarity or possibly even to conducting additional analyses to confirm that your observed patterns are biologically real. I hope that my wording will indicate to you where along this gradient each comment exists.

- We would like to thank the reviewer for their words of encouragement, and for their insightful and constructive comments.
- In light of the reviewer's concerns, we have made substantial revisions to the manuscript to strengthen our conclusions. In particular, we have provided more explicit detail about analytical decisions, greater consideration of alternative explanations and more in-depth interpretation of the findings. Specific details are given in response to the comments below.

Specific comments

Each of the comments below is connected to a specific line number or range of line numbers in the PDF version of your manuscript that I was given to review. My comments are as follows:

Line 74: "numerical dominance among species" is a phrase whose meaning is not immediately clear. Perhaps rephrase or expand; otherwise readers could mentally stumble when they encounter this phrase and wonder why the two patterns noted at the start of the sentence "suggest" anything.

- This wording was used to distinguish *species* dominance in a community (i.e. inequality in species abundance, the opposite of community evenness) from *behavioural* dominance, which has been the subject of research in a garden feeding context. We have adjusted the wording to make it clear that we are referring to the former (**L89-91**).

Line 76: What does "relative abundance" mean within the context of this sentence? I think that this should be defined in the main text. I do not think that you are meaning the sums of the maximum counts of the 5 most common species, but instead values from Fig. 2e; the y-axes of this figure panel is one type of relative abundance (i.e. as a proportion of the total number of birds summed across all

species), but there are other interpretations of what "relative abundance" means, including just the reported counts (where "relative" indicates that the counts are assumed to be some index of actual abundance). I have seen this latter meaning used frequently in bird monitoring literature.

- The reviewer's interpretation here is correct. However, to avoid any confusion, we have expressed the results differently to make it clear that we are referring to the cumulative abundance of ranked species, as a proportion of the total number of species in the community (L91-95).
- We have also edited the y-axis in **Figure 2e** and the description of the *k*-dominance analysis in the **Methods (L318-323)**.

Lines 87-89: Shouldn't the information in this last sentence be presented as the second last sentence, and the current second last sentence become last in this paragraph? I.e. the information in the last sentence is a pattern whose explanation/interpretation is given in the current second last sentence?

- We have expanded this section to include a more detailed explanation/interpretation of the findings, following the suggestion of reviewer #1 L87-89.

Lines 92-93: Possible alternative explanation: Spatial range expansion is bringing many wintering species in contact with a greater proportion of all feeder sites?

More generally, is there any geographic variation in the observed patterns. The current analyses are implicitly assuming that Great Britain can be treated as a uniform area.

- Thank you for this suggestion. We agree that the potential impacts of species range changes are worth considering as an alternative explanation for our findings. Therefore, we have compared distribution data from the 1981/82-83/84 Winter Atlas (Lack 1986) with equivalent data from Bird Atlas 2007-11 (Balmer et al 2013) to identify areas where a species had apparently colonised or apparently disappeared at a 10-km square resolution. Data have been assigned to each garden according to the 10-km square in which it is located. After averaging across 98% (n = 130) of all species represented in the GBFS dataset, we found that the median net change in the proportion of GBFS gardens within a species' range was only 2.49%. This suggests that spatial range expansion is unlikely to have influenced our measures of community composition. This new data exploration has been included in the **Results (L95-100)** and explained in the **Methods (L281-288)**.
- Regarding the more general effects of geographical variation – our study aims to examine the large-scale, long-term impacts of garden bird feeding and determine whether this can be linked to bird population changes at a national scale. Having provided unique evidence of the consequences of feeding on British bird communities in the broadest terms, this of course raises further questions about more nuanced patterns, such as those associated with geographical variation. However, it was beyond the scope of this paper to also address such questions here, as doing so would complicate and dilute the key messages being presented.

Line 99: Placed where it is, this parenthetical information from the phylogenetic regression could be interpreted as meaning that the statistics are describing the test of the phylogenetic effect...which I presume they are not. What about shifting the parenthetical test to immediately after "timeframe", a bit earlier in this sentence, to bring the statistics in direct contact with the statement of pattern that the statistics are supporting?

- Changed as suggested

Lines 102-103: Are these the regular/habitual feeder users? Or are these whatever assortment of 72 species for which population trends were available, some of which are and are not regular feeder users. Remind readers about this somewhere here.

- Edited for clarity.

Lines 109-111: I think that you've done yourselves a dis-service by adding this information almost as an afterthought to the analysis of population trends in human-dominated habitats. A comparison of population trends in human-dominated and "non-human landscapes" (that's a sort of awkward term, perhaps replace with "non-anthropogenic"...although that's not exactly right either in reference to pretty much anywhere in Great Britain) is the strongest evidence that you could produce that bird feeding is directly impacting bird species' population sizes.

I suggest that you frame this entire paragraph's contents as a formal contrast between population trends in human-dominated and non-human-dominated landscapes, and reanalyse your data in a way that would allow the human/non-human landscape contrast to be formally tested.

Of course, statistical significance in any such test would imply that you're overall conclusion would have to be nuanced a little bit, such that humans are having a dramatic effect on Great Britain's bird communities in places where humans are feeding bird (and not everywhere, as the current conclusion is suggesting).

- We disagree that a comparison of feeder-users' population trends between 'human-dominated' habitats and the 'wider countryside' would produce the strongest evidence that bird feeding is directly impacting species' population sizes. There are many habitat differences associated with urbanisation that could drive such variations in trend, of which feeding is just one. The analytical challenge, therefore, is to separate and test the influence of feeding specifically.
- We have approached this by limiting our analysis to look at what is happening to bird populations in urban areas specifically, for this is where we know garden bird feeders are definitely accessible to all birds and therefore most likely to have population level impacts. We then asked the question, of all the species residing in urban areas, is there any difference in population trends for the species that we know use garden feeders, compared to those that, generally, don't? In doing this, we are therefore able to directly attribute population changes to the use of garden bird feeders, and not some other feature of the urban environment. We believe this is the best possible test of the consequences of garden bird feeding on national populations, and as such, we have not reanalysed the data using the approach suggested by the reviewer. But in light of their comment, we realise that we have not been explicit enough in explaining the rationale and the value of the analysis being presented. We have edited this section of the results to address this (L157-169).
- The added value of then also testing the difference in population trends for the same suite of species across all other habitats in which they reside (i.e. the wider countryside) is that this then allows us to conclude whether or not the use of feeders in urban areas is, effectively, a likely driving force in the overall population changes observed nationally, rather than the alternative possibility, that birds have increased their feeder use due to density-dependence

(i.e. birds moving into gardens in response to the wider countryside becoming saturated). We have also now made this much clearer (**L170-178**).

- More generally, our study began by investigating what impact feeding has had on bird communities, and then asking “so what does this mean for bird populations more generally?” in terms of the effects found. It was not attempting to explain differences in trends among urban birds or between urban and rural. Therefore we have not followed the reviewer’s suggestion to reframe the paragraph as a formal contrast along these lines. However, we believe the changes we have made to this section of the manuscript, with the aid of reviewer #1 and reviewer #2’s comments, have helped greatly in supporting and strengthening the overall conclusions.

Line 172: Does this mean that migratory as well as resident species are included, given that 20 or 26 weeks would create a span of time from late fall migration to early spring migration? I think that readers should be told this in the text of the paper.

- Yes, the dataset does include migrants. The **Methods** state: “...*unusual visitors to garden feeders such as scarce migrants, summer migrants (recorded at the beginning or end of the winter period) and species not traditionally considered as garden visitors (e.g. wetland birds) have been retained...*”

Line 181: This data-filtering decision also allows for changes in migration dates of bird species through time to affect the measured changes through time in community structure. Is this a problem/source of bias?

- This is an interesting observation. Theoretically, temporal changes in the arrival and departure of summer migrant breeders, causing them to overlap with the GBFS survey period, have the greatest potential to cause bias in community measures. Our species list (Supplementary Table 3) includes only 15 summer migrants, all of which can also occur on passage when migrating north and/or south of the UK. Newson *et al* (2016, *Ibis* 158: 481-495) included 10 of these species in their study of long-term changes in UK breeding bird migration. Comparisons of the median arrival and departure dates for 1966 and 2010 indicate that migratory changes are unlikely to have altered the inclusion of these species in GBFS over time (Newson *et al* 2016).
- Other changes could have also occurred among partial migrants and winter immigrants, possibly adding noise to the data. However, overall, changes in migration dates seem unlikely to be a source of any temporal bias. However, we have described this, and acknowledged the issues arising, with reference to Newson *et al* (2016), in the **Methods** (**L257-260**).

Line 185-186: Is the maximum affected by not just how many weeks had data at a site, but which weeks?

The implicit assumption of saturation/asymptoting counts with the 20 weeks of effort seems reasonable to me, but it might still be good to mention (and maybe for a few species test) that this assumption is valid.

- We now verify the assumption that 20 weeks of survey effort is sufficient to measure both species richness *and* bird numbers accurately in the **Methods** (**L262-271**).

- We tested for a significant effect of survey effort on bird numbers for all 133 species in the data, using a GLMM of maximum count as a function of ‘weeks surveyed’ while controlling for site and year as random effects. Having adjusted p-values to account for multiple testing, we found that maximum counts were unaffected by survey effort for 94.0% of species. A one-proportion z-test confirms that this is significantly more than expected by chance ($\chi^2 = 101.2$, $p < 0.001$, 95% CI = 89.2 – 100.0%).
- We have also now noted that sites were surveyed in 25.13 weeks on average per winter in the **Methods (L263)**, further validating our data selection methods. The median number of weeks surveyed was 26. Therefore, although we included all sites with ≥ 20 weeks per winter, in the majority of cases survey effort was at, or close to, 100%. There was, therefore, insufficient variation in the number of weeks with survey data for the source of bias that concerns the reviewer to occur.

Lines 188-189: Hmm...as written I do not think that this statement is true. You are writing that one can directly compare the relative abundance values from species to species, but this assumes that an equal proportion of individuals within each species around a feeder site was not counted of each species. This assumption of equal detectability, and its recognized falsehood, is what motivated development of survey methods like occupancy modelling and distance sampling, so the inappropriateness of the assumption is widely recognized.

It does seem a lot more reasonable to me that one could compare the changes, from beginning to end of the study period, in relative abundance values across species, which is what I presume you actually want to do.

- We agree with the reviewer, and have changed “comparing relative abundances” to “comparing changes in relative abundance”.

Line 191: This isn't a true statement: you calculated 3 indices. Figure 2e shows an evenness statistic. Phrased another way, should the "Bird community indices" and "National-scale trends in bird communities" text be in closer proximity, and at the current location of the latter sub-section of the Methods?

- These sections of the **Methods** have been combined and rephrased for clarity (**L312**).

Line 195: Would it be useful to mention that a comparison of richness and Simpson's index values is informative, e.g., if richness is unchanged and diversity increases, then one can infer that evenness increases?

- This point had been added to the **Methods**.

Line 202: Given that you used bootstrapping, you should also have generated estimates of confidence for each year. Did statistical confidence vary systematically from start to end of the survey (such that any assumption of equal variance through time for regression purposes would be inappropriate), or was the variation around estimated means asymmetrical (such that calculating and interpreting means would be inappropriate)?

- We have used a traditional random rarefaction method, which resamples without replacement to standardise sampling effort and enable valid community comparisons (see Gotelli & Colewell 2001, *Ecol Letts* 4: 379-391). Therefore the bootstrap procedure does not measure

statistical confidence as such. Moreover, because it uses sampling without replacement, the variance of rarefied communities is uninformative, since it relates to the rarefaction proportion rather than to the size of the sample. For this reason, 95% confidence intervals around the rarefied national community estimates were not calculated or included in Figure 2.

- This has now been made clear in the **Methods (L330-334)**.

Line 208: My own experience is that participant-based classification along a rural to urban gradient is inaccurate across larger geographic areas, because participants' perceptions of what constitutes "rural" will vary depending on where they live. Specifically, I have seen that people living near larger cities will consider areas as being rural, whereas someone in a less populated area would not classify these same sites as being rural. I am basing this on having compared participant classification with land cover information from remotely sensed sources.

- Survey participants are provided with clear definitions of "rural", "suburban" and "urban" sites for guidance, and classifications are validated by the survey organiser. So the influence of subjectivity on garden classifications should be limited.
- We have compared GBFS garden classifications with satellite-derived land cover data to verify their accuracy, as described below.
- There is no readily available data source spanning years 1973-2013 to reliably quantify the urban gradient for the whole survey period. Therefore, to best approximate true urban land cover for surveyed gardens, we have used Land Cover Map 1990 (www.ceh.ac.uk/services/land-cover-map-1990), since this was produced around the mid-point of our survey period. In support of the data, 'rural' gardens are located in 1-km squares with a significantly lower proportion of urban cover than 'suburban/urban' gardens ($n = 1006$, $\chi^2 = 40265$, $p < 0.001$). Specifically, according to LCM 1990, rural gardens had 12.04% (0.70 SE) total urban cover in the surrounding area on average, while suburban/urban gardens had 42.66% (1.06 SE). These land cover values can only be considered an approximation due to the temporal differences in the data collection, however, they do confirm that the habitat classifications used in this study are meaningful.
- We have now included more detail about garden classification and the LCM 1990 garden land cover statistics in the **Methods (L297-305)**.

Line 212: I think that the reason for your choice to use the average should be explained, at least in a sentence. I'm suggesting this because my immediate reaction upon reading this text is that bird communities or feeder use may not be influenced as much by average temperature as by more extreme weather events.

- Since the bird data come from whole winters, not just the aftermath of specific weather events, average winter temperature gives a more accurate representation of the conditions influencing our bird community metrics. This is now stated in the **Methods (L306-308)**.

Lines 226-228: Why did you choose to use this approach, and not fit a regression through all of the data? I am not suggesting that your approach is wrong, but I (and I think other readers) would like to know the reason for comparing beginning to end, without examining whether there has been a monotonic change through time. This assumption of monotonicity is implicit in the method that you have chosen, is it not?

- No, this method does not make any assumptions about the shape of the trend. It simply provides a reliable measure of *net change*, in the same way that national population changes are measured. Using regression lines would have produced a poor measure of change due to the highly non-linear temporal patterns of feeder use.

Given that new food types were introduced at different points in time, we might expect non-linear increases in feeder use through time to be seen. I am not suggesting that analyses such as these should form part of the material presented in the paper. However, I am wondering whether you have looked at the pattern of change in feeder use through time (e.g., fitting a species' data with a GAM to visualize the shape of the change in abundance), which would allow you to state in the manuscript that you have verified your implicit assumption that the changes in feeder use that you observed were monotonic.

- We agree, it would be interesting to look at patterns of change for all species using GAMs. However it was outside the scope of this paper to go into that degree of species-level detail. Nonetheless, we did discuss the approach being taken here at length. Specifically, we were interested in the associations between changing feeder use and wider changes in national population sizes. Long-term national population change is measured by comparing annual population indices at the beginning and end of a time series (see Massimino et al 2017 <https://www.bto.org/about-birds/birdtrends/2017>). We therefore aimed to produce a comparable measure of change for garden feeder use. We used three years of data from the beginning and end of our time series and a binomial GLM to estimate change in order to avoid introducing biases by only using data for the first and last years (see Massimino et al 2017 <https://www.bto.org/about-birds/birdtrends/2017>).
- We have now made the rationale for this approach clearer in the **Methods (L353-358)**.

Lines 229-231: I think that a bit more information would be useful, or at least interesting, for readers here. First, I presume that converting the raw data, which are in the form of maximum counts of birds at each in each year, into presence/absence data and thus lowering the information content of the data was deliberate. However, the reason is not explained to readers. Second, for the sake of clarity I think that the text should be altered slightly to make clearer that each site was represented up to 3 times in the "start" and 3 times in the "end" data. I presume that this was the case, but for the sake of allowing readers to fully understand and potentially replicate the analyses, your actual treatment of the data should be described just a bit more clearly.

- Unfortunately there appears to be some misunderstanding about the analysis being referred to. 'Feeder use', the response variable, was the proportion of GBFS gardens in which a species was observed using feeders each year. This uses data from all gardens to derive a single, intuitive index for measuring changes in the extent of species overall feeder use nationally and comparing this to population change. Using site occupancy to derive feeder use, as opposed to species abundance, minimises the influence from stochastic variation in counts, caused by ephemeral flocks, for example. Moreover, although this could be considered 'lowering the information content of the data', it produces an index that is more meaningful and easy to interpret (in terms of understanding the scale of feeder use) than a measure based on summed or averaged species counts would.

- Since this is a national- rather than garden-scale analysis, it is unnecessary to refer to the number of times ‘each site was represented’.
- We have revised this section of the **Methods** to hopefully make it much easier to understand (**L353-364**).

Lines 217-233: Both with the calculation of the change in feeder usage and the change in population index, you are using statistical modelling to create an estimate of something, and each of these estimates has an error associated with it. Then differences between estimates from the "start" and "end" periods are calculated, but the uncertainties from the original estimates are discarded. Finally, these two differences are regressed against each other and a p-value is calculated that ignores the uncertainty that was not carried through from the initial estimates. Minimally, I believe that this means that the statistical probability of association in the final regression is too small by some unknown amount, because sources of error were discarded in the processing pipeline. I think that minimally this deserved to be at least recognized in the paper's text. Additionally, is there any possible way that not propagating errors through the processing could cause the pattern that you observed as an artifact? I cannot see how this could happen, I will add, but I think that it is a question that you should consider.

- The reviewer is of course correct, there is a level of uncertainty around estimates of feeder use change and population change, since each estimate (feeder use and population index) for the beginning and end of the time series has an error associated with it. To account for this error, we have used a bootstrap procedure to produce 95% confidence limits around the feeder use change – population change regression line (**Figure 3b**), allowing all uncertainty across the modelling process to be accounted for within the final model outcome.
- For each bootstrap sample (n = 1000), new values of feeder use and population index for the beginning and end of the time series were drawn at random from the confidence limits around their original estimates, and then used to recalculate estimates of change. The PGLS model was fitted to each bootstrap sample with lambda set at the value estimated for the original model, then 95% confidence limits were calculated from the set of regression coefficients produced.
- Of particular note, the PGLS regression relationship was highly significant across all bootstrap samples and the resulting confidence region around the fitted line is small. We are therefore confident that the relationship reported is statistically sound and independent of any uncertainty around the population estimates.
- A description of the bootstrap procedure has been added to the **Methods (376-383)** and the 95% confidence limit is now shown in **Figure 3b**.

Lines 259-260: I'm not entirely sure, but I think that the refitting would not serve to allow statistical significance to be "better approximated", particularly if the GAMM "shrank" the smoothing trend to be a linear regression. For near-linear trends though, simplifying to a straight line would have the advantage that you would have a simple slope parameter that is easy to interpret and is automatically produced as output.

- Linear models were only fitted after checking for and discounting evidence of non-linearity, as already described in the **Statistical modelling** subsection (see **403-407**), since it is

disadvantageous to fit a non-linear model to a linear trend for the reasons the reviewer describes.

Lines 261-263: Where the spline correlograms fitted to the residuals from GAMMs and LMMs? I presume that this is the case, but you need to state which type of information was explored for spatial autocorrelation.

- Spline correlograms were fitted to the raw data; this is now stated (L399-401). Since there was no evidence of spatial autocorrelation within the raw data, it was not necessary to fit correlograms to model residuals.

Lines 280-283: You have explained what you have done (i.e. controlling for changes in magazine length, and the amount of advertising through time), but you have not explained why these corrections are necessary. I.e., there are implicit assumptions that I think should be stated.

To me, it looks like you are assuming that fewer types of bird food will be advertised if there is less space in which to advertise...but wouldn't it make sense for a company marketing a product to deliberately differentiate themselves by advertising a novel food item (i.e. the number of unique food items should be relatively independent of available space, at least beyond some amount of available space...suggesting non-linear relationships between available space and the number of unique food types)? So, could your statistical corrections possibly be resulting in an underestimate of the change through time if you are forcing linear corrections?

- The description of the garden bird feeding industry methodologies has been substantially reworked to improve clarity and to better explain the rationale behind controlling for advertising space (L215-240).
- Indeed, the assumption is that the number of products advertised will increase with greater advertising space availability. To test change in food/feeder numbers, all individual products attributed to each company are summed per magazine. By controlling for advertising space, this enables us to draw the conclusion that, effectively, companies are expanding their product ranges over time, and that the observed increases are not simply an artefact of the increase in advert size/number. We agree, the relationship between advertising space and number of products will be asymptotic. However, without knowing where the asymptote lies, we believe the analysis is more defensible with available advertising space accounted for.

Line 290: You described how there was a saturation in species richness at sites after there were at least 20 weeks of observations submitted for a site in a season. However, does the abundance (i.e. the maximum number of birds observed at a single time) also saturate at this point, or does abundance increase with the number of observation periods, beyond the minimal 20 observation periods, for any species?

I would be surprised if this was an important effect, but possibly something worth checking and mentioning in passing at the same time that the saturation in richness is mentioned on line 172?

- As discussed above, and now in the **Methods (L266-271)**, abundance did not increase with the number of weeks surveyed for the majority of species.

Lines 295-296: Why do you need to write "using the LMM modelling framework" when "using LMMs" says the same thing?

- This has been changed in the revised **Methods**.

Lines 303-304: Would direct comparability after standardizing be possible if the values of the predictor variables were not normally (or at least symmetrically) distributed around their mean? So, are you sure that the interpretation based on direct comparability is legitimate for your data, or could interpretations have been biased? I do not know the answer to my question, but this was a question that came to mind as I read this text.

- In regression, there are no assumptions about the distributions of predictor variables, so the interpretation of the effect of x on y is independent of how x is distributed. Therefore, there is no impact of the underlying predictor data distribution on the interpretation of their effects, irrespective of whether or not these data have been standardised. Standardisation simply puts the predictors on the same scale, so that the relative importance of the different predictors can be more easily interpreted.

Figure 3, panel A: So, increased feeder use means a greater diversity of food is being provided, thus attracting a greater diversity of bird species. At least, that's the presumption. Presumably then it should be the "non-traditional" feeder birds that benefitted most (and increased in population size the most). However, wouldn't that mean that the PGLS should show an effect of phylogeny because these non-traditional feeder birds should be more taxonomically similar? I.e., is there some biological meaning that can be obtained from the lack of a phylogenetic signal, and if so would it be worth mentioning in the paper?

- Figure 3a illustrates changes in the use of garden bird feeders between 1973 and 2012 for the 39 species we defined as 'feeder-users' (recorded in an average of $\leq 10\%$ of GBFS gardens per year). An increase in feeder use could arise through a number of mechanisms, in addition to the effects of greater food diversity. For example, innovation in feeder design, better quality of the same foods and behavioural adaptation could all influence the frequency of feeder use and the benefits accrued, with downstream consequences for population size (**now stated in the Discussion L201-204**). Therefore, it is difficult to predict which species are likely to be 'benefitted most'.
- Nonetheless, it seems reasonable to predict that more closely related species would be more similar in their tendency to adopt/use supplementary foods, and this was our assumption when we accounted for phylogenetic structure in our analyses. In reality, the taxonomic variation in feeder use appears to be more complicated. As the phylogeny in Supplementary Fig 4 shows, there is no distinct clustering of feeder-users, with use of supplementary food evident across the phylogenetic tree.
- We have added further discussion around the phylogeny results to the main text (**L146-156**).

Figure 3, panel A: Strictly speaking the wording on the x-axis is wrong, I think. The values plotted on the axis are not changes, but actual proportions of sites with species attending feeders in each of two years.

- The x-axis label for **Figure 3a** has been corrected.

Extended Data Table 3: I think that it would be useful to anyone scanning this table if the species that you define as feeder users were indicated (e.g., by putting their names in bold font). Otherwise, readers will have to cross-reference with another location to find this information.

- Species defined as feeder-users are now shown in bold in **Supplementary Table 3**.

Reviewer #1 (Remarks to the Author):

I am satisfied with the authors' changes to the manuscript in response to my comments, and pleased to see that they now present a much stronger, balanced, and robust narrative to support their study. The many implicit assumptions throughout the manuscript are now explicitly outlined and/or justified, and the writing flows considerably better, with added clarity and context so that the reader is not left guessing at the authors' intended meaning.

I have a few additional minor comments below on the revised text. With these points addressed, I believe the paper is ready for publication. I congratulate the authors on what will be a valuable addition to the field.

L40: "trajectories" – spelling

L51-52: "times series" – correct to "time series"

L61-63: the addition of the reference here is good, however, it is still an assumption that the advertising patterns of bird feeding products, specifically, are a reliable index of consumer behaviour – this particular relationship has not been verified. I suggest tweaking the wording of the sentence to reflect this – e.g. advertising patterns "are expected to provide meaningful indices," instead of "can provide reliable indices". E.g. you do not know how reliable your index is, but assume it to be meaningful nonetheless.

L159-165:

This is confusing. To rephrase your sentence, you are saying that to test the effect of feeding independently of the factors operating in urban environments, you chose to focus your analysis on urban areas? This does not make sense. You are testing within a selected habitat type to attempt to standardise these other factors, rather than being independent of them.

Also, you use a number of descriptions here (urbanised areas of Britain, vs. 'urban' areas of Britain, vs. urban environments). I suggest considering whether you can be more consistent throughout.

L168: "species that do not using feeders" – correct to "use"

L181: "Wild bird feeding has become engrained into human culture" - in some cultures, but not all.

L187-189: These sentences are improved, but still need work: You negate the previous sentence and studies by starting with 'however' – remove this and instead build on that previous sentence. You are not explicit as to what consequences you are referring, and it is grammatically incorrect to say "more far-reaching". I suggest the following edit: "Our findings indicate that the consequences of feeding reach further still, with evidence that this habitual human activity is associated with national-scale restructuring of bird communities."

L192-194: "While those showing reduced relative feeder use are failing to benefit as behaviourally dominant or better adapted species become more common within the community." This study does not investigate the underlying mechanisms for the changes – reword so that it is clearer this is speculation only.

L196: "Feeding may, therefore, already have had important effects" – I would say it is highly likely to have already had important effects.

L196-197: "and more coordinated activity" – highly ambiguous what you mean by this. Coordinated by whom? What activity? Clarify.

L197: "a fascinating opportunity" – fascinating doesn't seem like the most appropriate word here. Fascinating for whom? Scientists? The public?

L200: "As feeding intensifies, this will only exacerbate the..." – there is an inherent assumption here, that feeding will continue to intensify on the same trajectory. Better to say: "If feeding continues to intensify, it will likely exacerbate..." or something along those lines.

Reviewer #2 (Remarks to the Author):

I think that you have done a very careful and thoughtful job of revising your manuscript in light of my original review, and I am very happy to see that your story told by your paper is robust to the potential confounds or complications that I had suggested might be present. I am commenting on a few locations in your revised manuscript, but these comments are essentially all suggestions for potential slight changes in wording to make your communication of information even clearer. These locations, based on the line numbers in the PDF version of the revised manuscript that I reviewed, are as follows:

Line 145: Consider adding a very few words to the phrase “national population changes” in order to inform readers in the main body of the paper that these estimates of population change are essentially entirely independent of the data on garden bird feeding.

Line 235: Maybe explain to readers why you chose to use advertising pages as an offset and not a covariate. Also, and maybe more importantly, do you want to justify using the number of pages as a log-linear effect on the number of products (i.e. why is this a reasonable implicit assumption?).

Line 278: Is “standardized” the most appropriate word with which to describe how the effects of variation in sampling effort was controlled?

Line 294-296: Consider explaining in biological terms what is assumed by log-transforming the numbers of feeders. The relationship between log-transformed numbers of feeders and year in a LMM will be non-linear. If you describe the form of this non-linearity to readers, they will more easily be able to assess if this functional form of relationship is logical.

Lines 306-308: Here I am revisiting a comment that I made on the original manuscript. I think that by using average temperature as your climate predictor you are implicitly assuming that extreme weather events are not a major force affecting the distribution of birds in winter in Britain. Why not just tell readers that this is the biological assumption that you are making when you use mean monthly temperatures instead of leaving this assumption implicit? Note that I am not suggesting that your assumption is not valid, but merely suggesting that explicit assumptions are more appropriate than implicit assumptions.

Line 326-327: I think that “bootstrapping” is by definition “sampling with replacement”. Further how can there be “resampling without replacement”? I think that you need to replace the phrase “bootstrap-resampling without replacement” in order to accurately describe what you have done. Perhaps use “sub-sampling” or just “rarefaction” to describe the numeric processing?

Line 329: Again, the word “resampled” seem inaccurate to me, given that the sampling was done without replacement, while the “re” in “resampling” to me implies that sampling with replacement was conducted.

Line 330: What you have done here, averaging measurements to produce a single value per year, seems reasonable to me. Have you considered, though, whether there would be added information provided if you were to treat each of the 1000 rarified samples and a separate data point? I.e., does using a single average value underestimate the uncertainty in the fitted GAM?

Line 353-355: This is a second instance in which I'm revisiting a comment that I made on the original version of the manuscript. Again, I am not saying that I think that what you have done is incorrect. However, looking at the pattern of change through the years, using information from all years, could potentially reveal non-linearity in the pattern such as an increase in probability of feeder use through time followed by a decline at the very end...and such a pattern would run counter to your implied argument that there has been a continuous increase in feeder populations of feeder birds through time. I still think that you are implicitly assuming, but have failed to test, whether the change in feeder use has been monotonic. So, this is another instance in which you seem to be making an implicit assumption where it would be appropriate to make this assumption explicit to readers.

Line 398: Was there a substantial amount of variance accounted for by the temporal auto-regressive correlation structure? You tell readers that spatial autocorrelation was at unimportantly low levels, but you have not provided readers with an understanding of whether the AR(1) temporal correlation structure was reasonable/useful.

Reviewers' comments

Reviewer #1:

I am satisfied with the authors' changes to the manuscript in response to my comments, and pleased to see that they now present a much stronger, balanced, and robust narrative to support their study. The many implicit assumptions throughout the manuscript are now explicitly outlined and/or justified, and the writing flows considerably better, with added clarity and context so that the reader is not left guessing at the authors' intended meaning.

I have a few additional minor comments below on the revised text. With these points addressed, I believe the paper is ready for publication. I congratulate the authors on what will be a valuable addition to the field.

- We are extremely appreciative of the reviewer's positive comments. We are happy to learn that they are satisfied with the changes made to the original manuscript, and we thank them again for their constructive feedback.

L40: "tragectories" – spelling

- Corrected.

L51-52: "times series" – correct to "time series"

- Corrected.

L61-63: the addition of the reference here is good, however, it is still an assumption that the advertising patterns of bird feeding products, specifically, are a reliable index of consumer behaviour – this particular relationship has not been verified. I suggest tweaking the wording of the sentence to reflect this – e.g. advertising patterns "are expected to provide meaningful indices," instead of "can provide reliable indices". E.g. you do not know how reliable your index is, but assume it to be meaningful nonetheless.

- Changed as suggested.

L159-165:

This is confusing. To rephrase your sentence, you are saying that to test the effect of feeding independently of the factors operating in urban environments, you chose to focus your analysis on urban areas? This does not make sense. You are testing within a selected habitat type to attempt to standardise these other factors, rather than being independent of them.

- By way of explanation: we show a significant positive association between feeder use and species' population change for the whole of Britain. However, bird population trends are significantly different in urban habitats compared to other habitats (Sullivan et al 2015, Biol Conserv). Because the prevalence and use of feeders is likely to co-vary with other urban characteristics that might also influence garden bird numbers (e.g. disturbance, predation pressure, habitat composition etc), it is difficult to differentiate the effects of feeding on national population change from the other general effects of urbanisation. By focusing on

urban habitats specifically, we can confidently conclude that the differences in species population trends observed (Fig 3c) are unlikely to be a consequence of urbanisation *per se*.

- We have edited the text and added additional detail to avoid any further confusion (L160-167).

Also, you use a number of descriptions here (urbanised areas of Britain, vs. 'urban' areas of Britain, vs. urban environments). I suggest considering whether you can be more consistent throughout.

- The terminology has been revised. It now includes only 'urban'/'urban areas' (i.e. areas dominated by humans) and 'urbanisation' (i.e. the process of becoming more urban).

L168: "species that do not using feeders" – correct to "use"

- Corrected.

L181: "Wild bird feeding has become engrained into human culture" - in some cultures, but not all.

- Changed to "...human culture across many areas of the World" to make it clear that feeding is not universally undertaken (L187-189).

L187-189: These sentences are improved, but still need work: You negate the previous sentence and studies by starting with 'however' – remove this and instead build on that previous sentence. You are not explicit as to what consequences you are referring, and it is grammatically incorrect to say "more far-reaching". I suggest the following edit: "Our findings indicate that the consequences of feeding reach further still, with evidence that this habitual human activity is associated with national-scale restructuring of bird communities."

- Changed as suggested.

L192-194: "While those showing reduced relative feeder use are failing to benefit as behaviourally dominant or better adapted species become more common within the community." This study does not investigate the underlying mechanisms for the changes – reword so that it is clearer this is speculation only.

- This sentence has been reworded to include '...appear to have...' to make it clearer that it is speculative (L197-201).

L196: "Feeding may, therefore, already have had important effects" – I would say it is highly likely to have already had important effects.

- Changed to 'highly likely'.

L196-197: "and more coordinated activity" – highly ambiguous what you mean by this. Coordinated by whom? What activity? Clarify.

- This has been edited to remove ambiguity and improve clarity (L203-207).

L197: "a fascinating opportunity" – fascinating doesn't seem like the most appropriate word here. Fascinating for whom? Scientists? The public?

- Changed, see above response.

L200: “As feeding intensifies, this will only exacerbate the...” – there is an inherent assumption here, that feeding will continue to intensify on the same trajectory. Better to say: “If feeding continues to intensify, it will likely exacerbate...” or something along those lines.

- Changed as suggested.

Reviewer #2:

I think that you have done a very careful and thoughtful job of revising your manuscript in light of my original review, and I am very happy to see that your story told by your paper is robust to the potential confounds or complications that I had suggested might be present. I am commenting on a the material in a few locations in your revised manuscript, but these comments are essentially all suggestions for potential slight changes in wording to make your communication of information even clearer. These locations, based on the line numbers in the PDF version of the revised manuscript that I reviewed, are as follows:

- We would like to thank the reviewer for their positive comments in response to our revised manuscript. We appreciate the further helpful suggestions, which we have addressed in turn below.

Line 145: Consider adding a very few words to the phrase “national population changes” in order to inform readers in the main body of the paper that these estimates of population change are essentially entirely independent of the data on garden bird feeding.

- Edited as suggested.

Line 235: Maybe explain to readers why you chose to use advertising pages as an offset and not a covariate. Also, and maybe more importantly, do you want to justify using the number of pages as a log-linear effect on the number of products (i.e. why is this a reasonable implicit assumption?).

- We expected that the numbers of products advertised would be proportional to the total amount of advertising space available. Therefore the log of ‘advertising pages’ was include as an offset in order to account for the effects of differences in the unit of observation (i.e. the reported changes in total advertising space through time) on product counts.
- For clarity, we have now stated explicitly that advertising pages was log-transformed and added an explanation for its inclusion as an offset (L243-245).

Line 278: Is “standardized” the most appropriate word with which to describe how the effects of variation in sampling effort was controlled?

- ‘Standardised’ has been replaced with ‘controlled’.

Line 294-296: Consider explaining in biological terms what is assumed by log-transforming the numbers of feeders. The relationship between log-transformed numbers of feeders and year in a LMM will be non-linear. If you describe the form of this non-linearity to readers, they will more easily be able to assess if this functional form of relationship is logical.

- The feeder counts data were positively skewed. Therefore these response variables were log transformed in order to meet the assumptions of the LMMs, by normalising the model residuals. A significant log-linear relationship indicates a constant *percentage* increase in feeders over time.
- This is now explained in the Methods (L308-310). For ease of interpretation, we have also changed the y-axes in **Supplementary Figure 3** so they are on the scale of the raw data.

Lines 306-308: Here I am revisiting a comment that I made on the original manuscript. I think that by using average temperature as your climate predictor you are implicitly assuming that extreme weather events are not a major force affecting the distribution of birds in winter in Britain. Why not just tell readers that this is the biological assumption that you are making when you use mean monthly temperatures instead of leaving this assumption implicit? Note that I am not suggesting that your assumption is not valid, but merely suggesting that explicit assumptions are more appropriate than implicit assumptions.

- This assumption has now been stated explicitly (L320-323).

Line 326-327: I think that “bootstrapping” is by definition “sampling with replacement”. Further how can there be “resampling without replacement”? I think that you need to replace the phrase “bootstrap-resampling without replacement” in order to accurately describe what you have done. Perhaps use “sub-sampling” or just “rarefaction” to describe the numeric processing?

- The rarefaction description has been rephrased to improve accuracy and clarity (L340-345).

Line 329: Again, the word “resampled” seem inaccurate to me, given that the sampling was done without replacement, while the “re” in “resampling” to me implies that sampling with replacement was conducted.

- Resampling can be with or without replacement. We have clearly stated the approach used, and we are following the terminology used elsewhere in the literature to describe rarefaction (e.g. Gotelli & Colwell, *Ecol Letts*, 2001; Magurran 2004 *Measuring Biological Diversity*). On this basis, we see no need to change the wording.

Line 330: What you have done here, averaging measurements to produce a single value per year, seems reasonable to me. Have you considered, though, whether there would be added information provided if you were to treat each of the 1000 rarified samples and a separate data point? I.e., does using a single average value underestimate the uncertainty in the fitted GAM?

- For the same reason that rarefied samples should not be used to calculate confidence intervals (i.e. variance around the mean is dependent on the proportion of sites in the subsample and therefore doesn’t reflect uncertainty), as we described in response to the reviewer’s comment on the previous version, it would be incorrect to consider the 1000 replicates as separate data points.

Line 353-355: This is a second instance in which I’m revisiting a comment that I made on the original version of the manuscript. Again, I am not saying that I think that what you have done is incorrect. However, looking at the pattern of change through the years, using information from all years, could potentially reveal non-linearity in the pattern such as an increase in probability of feeder use through time followed by a decline at the very end...and such a pattern would run counter to your implied

argument that there has been a continuous increase in feeder populations of feeder birds through time. I still think that you are implicitly assuming, but have failed to test, whether the change in feeder use has been monotonic. So, this is another instance in which you seem to be making an implicit assumption where it would be appropriate to make this assumption explicit to readers.

- As we stated previously, we are not making any assumptions about the shape of the relationship. We are estimating net, overall change, analogous to the measures of population change we later use for comparison. A net increase/decrease simply implies that food use is greater/lower at the end of the time-series compared to the beginning. This measure is independent of the shape of the relationship year-on-year, and therefore it does not require the relationship to be monotonic.
- To avoid any further confusion, we have stated explicitly that we are not making any assumptions about the shape of the trend (**L372-375**).

Line 398: Was there a substantial amount of variance accounted for by the temporal auto-regressive correlation structure? You tell readers that spatial autocorrelation was at unimportantly low levels, but you have not provided readers with an understanding of whether the AR(1) temporal correlation structure was reasonable/useful.

- Evidence of temporal autocorrelation was detected by comparing models with and without an autocorrelation structure using AIC and by visual examination of auto-correlation plots for the model residuals. The AR(1) correlation structure was found to be optimal for accounting for temporal autocorrelation across multiple response variables and model types (i.e. linear / non-linear).
- This is now clearly stated (**L417-420**).